# NEURAL SLOT INTERPRETERS: GROUNDING OBJECT SEMANTICS IN EMERGENT SLOT REPRESENTATIONS

## ABSTRACT

Several accounts of human cognition posit that our intelligence is rooted in our ability to *form* abstract composable concepts, *ground* them in our environment, and *reason* over these grounded entities. This trifecta of human thought has remained elusive in modern intelligent machines. In this work, we investigate whether slot representations extracted from visual scenes serve as appropriate compositional abstractions for grounding and reasoning. We present the Neural Slot Interpreter (NSI), which learns to ground object semantics in slots. At the core of NSI is an *XML*-like schema that uses simple syntax rules to organize the object semantics of a scene into object-centric schema primitives. Then, the NSI metric learns to ground primitives into slots through a structured objective that reasons over the intermodal alignment. We show that the grounded slots surpass unsupervised slots in real-world object discovery and scale with scene complexity. Experiments with a bi-modal object-property and scene retrieval task demonstrate the grounding efficacy and interpretability of correspondences learned by NSI. Finally, we investigate the reasoning abilities of the grounded slots. Vision Transformers trained on grounding-aware NSI tokenizers using as few as ten tokens outperform patch-based tokens on challenging few-shot classification tasks.

## 1 INTRODUCTION

Humans possess a repertoire of strong structural biases, a kind of abstract knowledge that enables us to perceive and rapidly adapt to our environments (Griffiths et al., 2010). *Compositionality* is one such structural prior that helps us systematically reason about complex stimuli as a whole by recursively reasoning about its parts (Zuberbühler, 2019; Lake & Baroni, 2023). We decompose broad motor skills into finer dexterous finger movements, sentences into words and phrases, and speech into phonemes. In the visual world, the concept of "objectness" serves as a natural compositional prior, enabling us to decompose novel scenes into familiar objects and reason about their properties (Lake et al., 2016). We also have the uncanny ability to connect real-world entities and concepts to these abstract object-like symbols in our heads, canonically referred to as the *grounding problem* (Harnad, 1990; Greff et al., 2020). For instance, human infants, while looking at a zebra for the first time, might excitedly conclude that it is, in fact, "a striped horse." *If grounded object-like representations are fundamental to human-like compositional generalization, how do we instill these inductive biases into neural network representations?*

Unsupervised object-centric autoencoder models (Burgess et al., 2019; Greff et al., 2019; 2020; Locatello et al., 2020; Engelcke et al., 2019; 2021; Singh et al., 2021; Chang et al., 2023b; Seitzer et al., 2023; Kori et al., 2024) have become increasingly adept at learning object-centric representations called *slots* from raw visual stimuli. Further work has demonstrated that learned slots can be flexibly composed together for tasks like scene composition, causal induction, learning intuitive physics, dynamics simulation, and control (Dedhia et al., 2023; Jiang et al., 2023; Wu et al., 2023a;b; Chang et al., 2023a; Jabri et al., 2023). While object slots hold promise as a compositional building block for machines that mimic human abstraction and generalization, a key challenge emerges. Unlike humans, these learned slots lack grounding in real-world concepts. For example, a slot representation of an object like "apple" could refer to the fruit, the company, or a generic round artifact. Without grounding, a slot-based system cannot disambiguate these meanings effectively (Haugeland, 1985) and is fundamentally limited in its embodied reasoning abilities. Prior works (Locatello et al., 2020; Seitzer et al., 2023; Kori et al., 2024) have tackled learning to ground slots by predicting object

properties (texture, material, category, etc.) from the representations. The grounding objective is, therefore, *implicit* within the prediction of object semantics. However, ground truth correspondences between object concepts and slots are generally unknown, restricting prediction to a set-matching template. Under a set-matching framework, a *single* slot predicts object properties of a *single* object, thereby constraining the grounding information assimilated per slot. We circumvent the limitations of prediction as a surrogate for grounding by making the grounding objective *explicit* in the form of a co-training paradigm that we call the *Neural Slot Interpreter* (NSI).

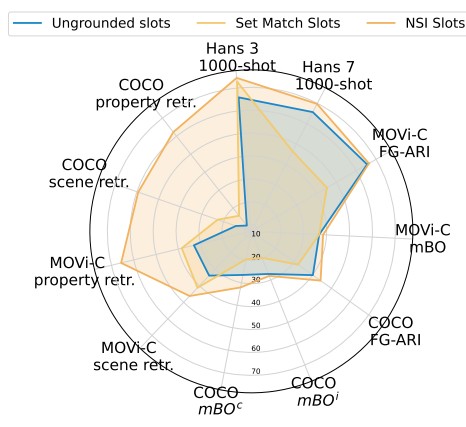

Figure 1: NSI abstracts grounded slots from scenes and enhances object discovery, grounding efficacy, and downstream reasoning abilities of slot representations.

The core insight behind NSI is simple: instead of *predicting* a single object-concept from a slot, *assign* multiple concepts to slots over a shared latent space. Our primary contribution is a similarity metric that explicitly reasons about the intermodal assignments. Notably, the proposed metric for NSI supplants the one-object-per-slot assumption and facilitates flexible assignment. Contrastive learning over the similarity objective yields grounded slots that outperform their ungrounded and set-matched counterparts over a broad swath of tasks (see Fig. 1). We propose an object-centric annotation schema in Section 3.1 for dense alignment to organize scene annotations for grounding into slots. We describe the design of a hierarchical transformer-based architecture in Section 3.2.2 to extract neural representations from the schema. To enable slots to ground a wide array of concepts flexibly without relying on matching templates, we formulate a bi-level scoring metric over a learned latent space in Section 3.2.

*Do NSI modules require specialized training recipes?* Our experiments demonstrate that NSI is a plug-and-play paradigm that can be easily augmented to the traditional object-centric learning objective. *Does NSI preserve compositionality?* NSI preserves and often improves visual compositionality, as shown in Section 4.2, where we demonstrate its competitiveness on object discovery benchmarks. *Are notions of objects effectively grounded in emergent slot representations?* We validate the efficacy of grounded slots on a bimodal property-image retrieval task in Section 4.3, where we show that NSI surpasses the state of the art. *Are NSI-grounded slots effective substrates for visual reasoning?* In Section 4.4, we train a vision transformer (ViT) (Dosovitskiy et al., 2021) for a scene classification task where significantly fewer grounded slot tokens show improved performance and adaptability over traditional patch-based tokens. *Can dense associations learned by NSI inform real-world reasoning systems?* Our experiments described in Appendix F.1 show the usefulness of learned correspondences in identifying and locating objects in diverse scenes. Concretely, our contributions are as follows:

1. We present NSI, a co-training grounding paradigm for object-centric learners. We also propose an object-centric annotation schema called *visXML* for dense slot-label alignment. We formulate a similarity metric that measures scene-schema similarity by recursively reasoning over the similarity of compositional attributes of the respective modalities. NSI utilizes the metric to ground slots via a contrastive learning objective.

2. Our experiments demonstrate the efficacy of NSI over a wide array of tasks that encompass (1) object discovery, (2) scene-property retrieval, (3) few-shot scene classification, and (4) object detection. Overall, we find that grounded slot representations are key to object-centric perception, property grounding, and downstream adaptability for object-centric reasoning.

## 2 RELATED WORK

**Object-Centric Learning.** Researchers have formulated inductive biases for learning composable visual representations called 'slots' from raw visual stimuli (Burgess et al., 2019; Greff et al., 2019; Locatello et al., 2020; Greff et al., 2020; Engelcke et al., 2019; 2021; Singh et al., 2021; Seitzer et al., 2023) and auxiliary temporal information (Kipf et al., 2021; Elsayed et al., 2022; Singh et al., 2022). While this line of work demonstrates unsupervised object discovery, the adoption of slot

representations for grounding scenes remains largely underexplored. Prior works have been limited to using the Hungarian Matching Criterion to align slots to ground-truth property labels for property prediction (Locatello et al., 2020) or fine-tuning shallow property predictors on pre-trained backbones (Seitzer et al., 2023). A recent work (Kori et al., 2024) improves the predictive power of slots by learning quantized priors. Fundamentally, the single-object prediction per slot constraint poses the difficult problem of learning highly specialized representations. In contrast, this work proposes an approach that explicitly reasons about grounding via a flexible assignment metric.

**Visual Tokenizers.** Patch-based tokens have been adopted as the standard for visual understanding (Dosovitskiy et al., 2021) and generation (Peebles & Xie, 2023). Variations of this template include discretized patch tokens (Du et al., 2024), mixed-resolution patch tokens (Ronen et al., 2023), and pruned patch tokens (Kong et al., 2022; Tang et al., 2023). Beyond patches, recent works have explored region-based tokens (Ma et al., 2024). However, these tokenizers are inherently grounding-agnostic, in contrast to humans, who possess the ability to abstract concepts based on linguistic or cultural grounding priors, (Segall et al., 1966; Winawer et al., 2007). To this end, our work explores a grounding-aware tokenizer.

**Program Induction.** Such problems occur in many guises across computer vision (Li et al., 2020; Wu et al., 2017), natural language processing (Xu et al., 2018; Devlin et al., 2017), and cognitive science (Ellis et al., 2020). Grounding slots in an annotation schema via NSI is akin to neural program induction in the schema space.

## 3 NEURAL SLOT INTERPRETERS

Recall that the goal of NSI is to ground concepts into slot representations such that the objects contained within the slot align with the embodied notions of the object. We begin by discussing our proposed Extensible Markup Language (*XML*) for annotating object concepts.

### 3.1 *visXML* OBJECT-CENTRIC ANNOTATION SCHEMA

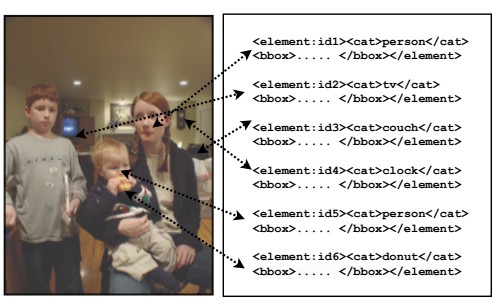

*visXML* is a simple markup syntax for abstracting scene labels as a collection of objects and their associated properties. Instances in *visXML* comprise multiple object-centric *primitives*. Primitives, in turn, are instantiated with respective object properties that form atomic units of the language. Each primitive starts with an `<element>` tag that identifies a unique object in the scene. The $<p_j>$ tags form the children of the `<element>` tags and capture properties $p_1, \cdots, p_J$ of the parent object. Some examples of $<p_j>$ tags are, but not limited to, shape, material, category, and object position. Thus, instance primitives naturally capture the notion of an object and neural representations extracted from primitives are, as such, well suited for being grounded in slots. In Section 4.1, we demonstrate the straightforward application of *visXML* organization on popular datasets. See Fig. 2 for an example instance and Appendix B.3 for more examples. On a more practical note, such markup languages are commonly-used software abstractions and can be ubiquitously interfaced with graphics engines, web APIs, or even large language models for semantic understanding (Dunn et al., 2022; Bubeck et al., 2023).

Figure 2: *visXML* description of a real-world scene. The dotted arrows show correspondences between primitives and the objects they annotate.

### 3.2 NSI GROUNDING TECHNIQUE

Scenes and their corresponding *visXML* instances capture object-centric representations through slots and primitives, respectively. NSI learns to align the object-centric representations of the respective modalities, i.e., slots and primitives, by grounding neural representations of schema primitives into slots (see Fig. 3). The grounding is learned by optimizing a contrastive learning objective over ground-truth scene-schema pairs (see Fig. 4). We describe the scene and schema encoder next.

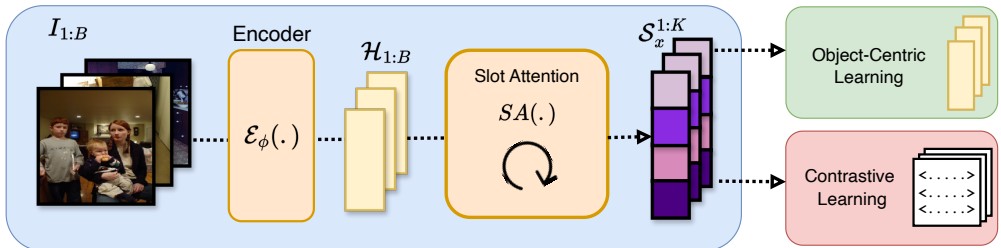

Figure 3: NSI Architecture. NSI augments object-centric learning autoencoders with a contrastive learning objective over a batch of scene-schema pairs. A DINOSAUR backbone (Seitzer et al., 2023) extracts slot representations $\mathcal{S}_x^{1:K}$ from a batch of scenes and a schema encoder extracts neural primitives $\mathcal{Z}_y^{1:N}$ from their corresponding schema pair. The slots are then passed to a decoder for reconstruction and the slot-primitive neural pairs are passed to the contrastive learning objective.

### 3.2.1 LEARNING SCENE REPRESENTATIONS

A given scene $I_x$ is represented via $K$ slots $S_x^{1:K} \in \mathbb{R}^{K \times d}$ abstracted from its perceptual features $H_x \in \mathbb{R}^{L \times c}$. For a given feature extractor $\mathcal{E}_\phi(.)$, the slots are obtained as

$$H_x = \mathcal{E}_\phi(I_x) \quad \Rightarrow \quad S_x^{1:K} = SA(H_x) \rightarrow \text{Slot Attention} \tag{1}$$

A spatial broadcast decoder $\mathcal{D}_\theta(.)$ (Locatello et al., 2020) reconstructs the features from slots, with the reconstruction error used as a learning signal:

$$\hat{H}_x = \mathcal{D}_\theta\left(S_x^{1:K}\right) \quad \Rightarrow \quad \mathcal{L}_{recon} = \left\| H_x - \hat{H}_x \right\|^2 \tag{2}$$

### 3.2.2 LEARNING SCHEMA REPRESENTATIONS

A bi-level architecture (see Appendix C.2) learns neural representations of the *visXML* primitives. First, a lower-level primitive encoder learns property-specific dictionaries $D(.)$ and embeds the property features into neural primitive representations. For discrete-valued properties, $D(.)$ is modeled as a simple lookup table of learnable weights, while continuous-valued properties are embedded via multi-layered perceptrons (MLPs). Let the dictionary $D_j(.)$ learn features for property $p_j$. Then, a primitive embedding $Z_{prim}$ is computed as:

$$Z = concat\left[D_1(p_1), \cdots, D_J(p_J)\right] \quad \Rightarrow \quad Z_{prim} = MLP(Z) \tag{3}$$

Note that these lower-level representations are schema-agnostic and only capture object-specific features. Then an upper-level schema encoder uses a bidirectional schema Transformer to further embed primitives, endowing representations with the overall schema context. For a given schema instance $P_y$ with $N$ primitives, the final representations $Z_y^{1:N} \in \mathbb{R}^{N \times d}$ are computed via Transformer $\mathcal{T}_{schema}(.)$ as:

$$Z_y^{1:N} = \mathcal{T}_{schema}\left(Z_{prim}^1, \cdots, Z_{prim}^N\right) \tag{4}$$

### 3.2.3 COMPOSITIONAL SCORE AGGREGATION

Recall that we want to ground entities $Z_y^{1:N}$ into the entities $S_x^{1:K}$. As a first step, we project these embeddings into a shared semantic space $Y \in \mathbb{R}^{d_{proj}}$. The projection head $\mathcal{H}_{scene}(.)$ for slots is modeled as the following residual network:

$$\tilde{Y}_x^k = W_{proj} S_x^k, \ W \in \mathbb{R}^{d_{proj} \times d} \quad \Rightarrow \quad Y_x^k = \tilde{Y}_x^k + MLP\left(LayerNorm\left(\tilde{Y}_x^k\right)\right) \tag{5}$$

Here, $LayerNorm$ denotes the layer normalization operation. A separate residual head $\mathcal{H}_{schema}(.)$ projects the primitive representations $Z_y^{1:N}$ into the semantic embeddings $Y_y^{1:N}$. Next, we supplant the traditional single object per slot assumption by assigning each primitive to its nearest slot in the latent space, as measured by dot-product similarity. The similarity score $\mathcal{S}_{xy}$ between a scene $x$ and primitive $y$ is the sum of nearest-neighbor similarities resulting from the primitive-slot assignment.

$$k_n^* = \underset{k \in \{1, \cdots, K\}}{\arg\max} \ Y_x^{k^T} Y_y^n \quad \Rightarrow \quad \mathcal{S}_{xy} = \sum_{n \in 1 \cdots N} \max_{k \in \{1, \cdots, K\}} \left(Y_x^{k^T} Y_y^n\right) \tag{6}$$

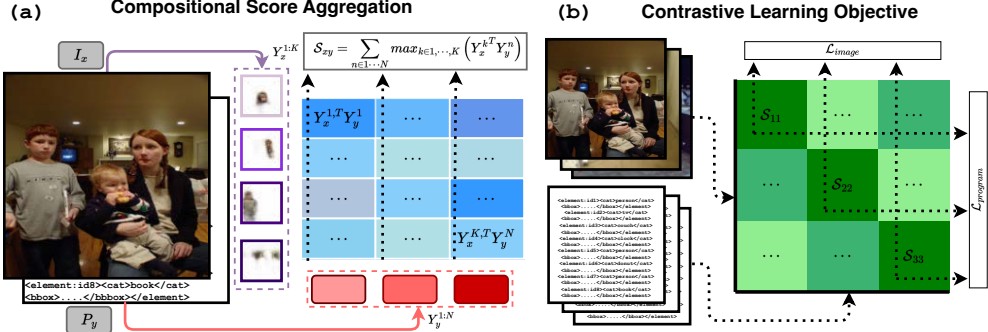

Figure 4: NSI similarity metric. (a) The inner loop of the metric computes the score $S_{xy}$ between compositional abstractions of an image $I_x$ and a schema $P_y$. Object slots and schema primitives are projected onto a shared embedding space and every latent primitive is assigned to its nearest slot for score aggregation. (b) The $S_{xy}$ scores obtained from *local* entities are used to optimize a *global* contrastive learning objective in the outer loop over a batch of image-schema pairs.

### 3.2.4 CONTRASTIVE LEARNING OBJECTIVE

The modality-specific embeddings and the resultant grounding are learned by optimizing a contrastive learning objective. More precisely, given a $B$-sized batch of {scene, schema} pairs, we use the $S_{xy}$ scores to distinguish the $B$ correct pairs from the $B^2 - B$ incorrect pairs. The probability of correctly classifying schema $P_x$ as the true pairing for scheme $I_x$ (and conversely predicting $I_x$ from $P_x$) is formulated as follows:

$$\mathbb{P}_x^{schema} = \frac{exp\left(S_{xx}/\tau\right)}{\sum_{y\in\{1,\cdots,B\}} exp\left(S_{xy}/\tau\right)} \quad \Rightarrow \quad \mathbb{P}_x^{scene} = \frac{exp\left(S_{xx}/\tau\right)}{\sum_{y\in\{1,\cdots,B\}} exp\left(S_{yx}/\tau\right)} \quad (7)$$

Here, the calculated scores are interpreted as logits and $\tau$ denotes the temperature parameter. The cross-entropy losses for scene and schema prediction are given by:

$$\mathcal{L}_{schema} = -\sum_{x\in\{1,\cdots,B\}} log\left(\mathbb{P}_x^{schema}\right); \mathcal{L}_{scene} = -\sum_{x\in\{1,\cdots,B\}} log\left(\mathbb{P}_x^{scene}\right) \quad (8)$$

The global contrastive learning objective is based on a symmetric cross-entropy (CE) loss as follows:

$$\mathcal{L}_{contrastive} = (\mathcal{L}_{scene} + \mathcal{L}_{schema})/2 \quad (9)$$

The overall training objective for NSI is given by:

$$\mathcal{L}_{train} = \beta_1 \times \mathcal{L}_{contrastive} + \beta_2 \times \mathcal{L}_{recon} \quad (10)$$

Note that $\beta_1 = 0.0, \beta_2 = 1.0$ corresponds to traditional autoencoder object-centric learning frameworks. Appendix C.3 presents the NSI pseudocode.

## 4 EXPERIMENTS

In this section, we set up experiments to answer (1) whether the NSI objective leads to the emergence of slots that bind to raw object features, (2) whether the slots are concurrently and coherently imbued into the notion of the object properties, and (3) if the slots are effectively grounded, do they enable improved reasoning over objects.

### 4.1 *visXML* INSTANTIATION AND ARCHITECTURE BACKBONE

Our experiments encompass different tasks on scenes ranging from synthetic renderings to in-the-wild scenes viz. (1) CLEVr Hans (Stammer et al., 2020): objects scattered on a plane, (2) CLEVrTex (Karazija et al., 2021): textured objects placed on textured backgrounds (3) MOVi-C (Greff et al., 2022): photorealistic objects on real-world surfaces, and (4) MS-COCO 2017 (Lin et al., 2015): a large-scale object detection dataset containing real-world images. In a pre-processing step, we organize scene labels into the *visXML* schema (Section 3.1). The property tags $<p_j>$ that populate

| Dataset | Property tags $<p_j>$ |
|---------|----------------------|
| CLEVr Hans | `<color>,<shape>,<material>,<size>,<3D position>` |
| CLEVrTex | `<texture>,<shape>,<size>,<3D position>` |
| MOVi-C | `<category>,<scale>,<2D position>,<bounding box>` |
| MS-COCO 2017 | `<category>,<bounding box>` |

Table 1: Property tags used to instantiate and ground schema primitives.

schema primitives in each dataset are listed in Table 1. We use the schema instances to ground object information in their corresponding scenes via NSI. See Appendix B.3 for *visXML* instances.

We followed the DINOSAUR (Seitzer et al., 2023) recipe for learning slot representations. DINOSAUR uses semantically-informative DINO ViT (Caron et al., 2021) features as an autoencoding objective, significantly improving real-world object-centric learning abilities. More specifically, we train an MLP decoder to reconstruct features from slots for all our experiments. Training details and hyperparameters are given in Appendix D.2.

## 4.2 OBJECT DISCOVERY

Object-centric frameworks have been traditionally used to bind neural network representations to distinct objects within a scene. Here, we evaluate whether grounded slots are more adept at discovering objects in the context of visual segmentation. To systematically probe the effect of grounding, we use a common backbone trained within the same compute budget and ablated across the grounding continuum: (a) **ungrounded slots** derived from the autoencoder objective, (b) set-matching prediction via Hungarian Matching Criterion (**HMC**), and (c) latent semantic assignment via **NSI**.

For each method, we extracted object masks from the backbone derived from slot-attention clusters and reported two segmentation metrics in Fig. 5(a): (1) Foreground Adjusted Rand Index (**FG-ARI**): measures the accuracy of clustering foreground objects into their respective segments and (2) Mean Best Overlap (**mBO**): assesses the best overlap between predicted and ground truth object masks. We report both instance ($mBO^i$) and class ($mBO^c$) level $mBO$ scores for COCO. Mask instances are shown in Fig. 5(b). We observed that:

(a) **NSI endowed slots meaningfully segment objects.** Object masks generated by NSI are competitive on synthetic scenes and markedly improve object discovery on COCO scenes. We posit that contrastive learning via NSI enhances symmetry breaking of the slot attention backbone for challenging real-world scenes, effectively binding slots to raw visual features.

(b) **HMC matching obscures object discovery on COCO.** Constraining slots to predict a single object forces the backbone to develop specialized representations for each object. In real-world scenes, this imposes a difficult learning problem and causes slots to deteriorate.

(c) **NSI is biased towards semantic classes over instance classes.** On COCO instances, semantic segmentation scores ($mBO^c$) are higher compared to the baseline. We attribute this to the NSI metric that biases slots to represent broader categories by grounding multiple objects.

## 4.3 GROUNDED COMPOSITIONAL SEMANTICS

Grounded concepts should be effectively aligned to the slots they represent. To this end, we set up a bimodal scene-property retrieval task that evaluates the grounding efficacy of various methods, i.e., the degree to which they learn to associate object properties with their representations. In the first half, models retrieve a set of object properties from a database, given the scene, using their respective alignment scores. In the second half, the task is inverted, where the models search for scenes, given the object properties. We use the test split across datasets as a retrieval database. For NSI, we use similarity scores $\mathcal{S}_{xy}$ as the retrieval metric and also evaluate various comparisons.

(a) **CLIP embeddings:** Such embeddings (Radford et al., 2021) form a strong non-slot baseline. Here, we encode the schema as a string and measure the similarity between the text-image CLIP embeddings.

(b) **Ungrounded slots** (Seitzer et al., 2023): We freeze the backbone and fine-tune a shallow predictor on it for property prediction using HMC (Kuhn, 1955). Subsequently, we use HMC scores for retrieval.

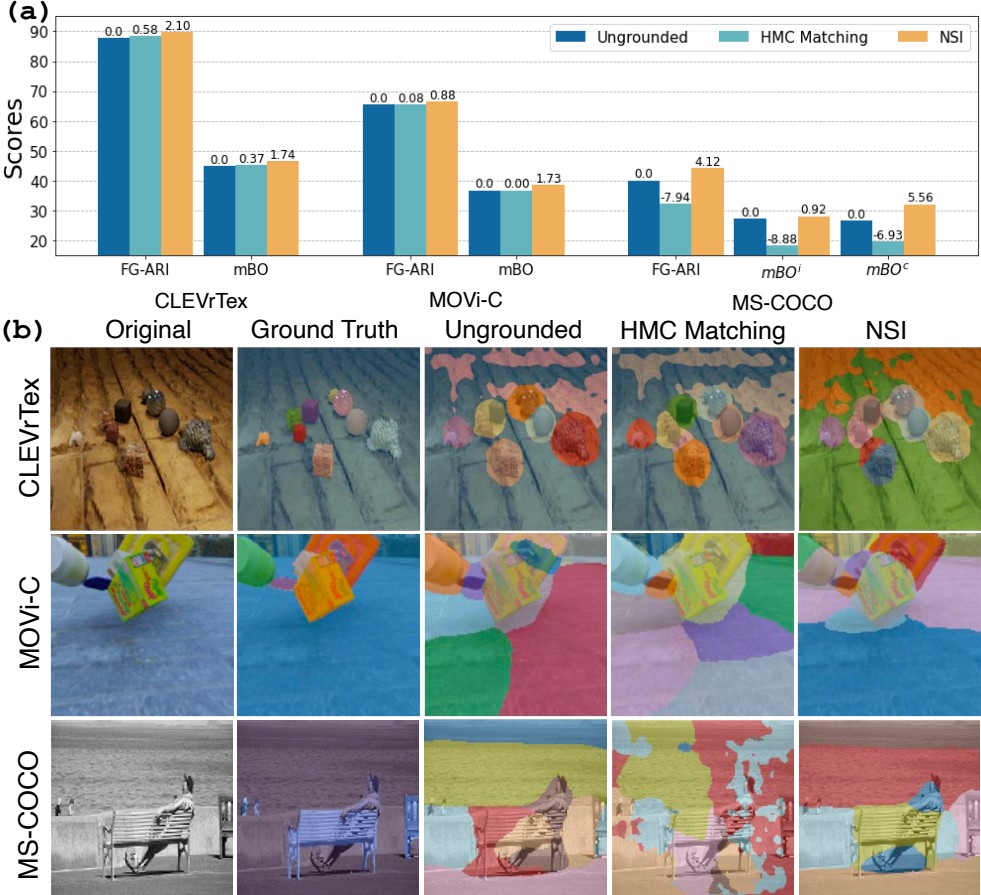

Figure 5: Object discovery results. (a) Segmentation results on three datasets. We report scores of $FG\text{-}ARI$ (higher is better) and $mBO$ (higher is better). We also show scores relative to the ungrounded baseline on top of each bar. (b) Visualization of attention masks learned by different models on instances of the datasets. See Appendix E.1.1 for error bars and more results.

(c) **HMC matching:** We fine-tune the slot architecture end-to-end to predict object properties from slots. We use the optimal HMC scores of fine-tuned slots for retrieval.

(d) **Ablations:** On NSI, where **NSI-ResNet 34** replaces DINO ViT with the ResNet backbone, as described in (Elsayed et al., 2022), and on **NSI-Schema Agnostic** where schema primitives are encoded without the schema Transformer (Appendix C.2).

In Fig. 6(a),(b),(c), we report Recall@(1/5), which denotes the accuracy with which the correct scene (property) is among the top (1/5) retrieved entities.

(a) **Grounded slots via NSI significantly improve semantic alignment compared to set matching:** NSI and its ablations that explicitly reason about compositional semantics enable improved retrieval compared to slots that rely on set-matching prediction. The performance gap widens as we evaluate the methods on more realistic datasets like MOVi-C and COCO.

(b) **The full model is essential:** Schema-agnostic encoders that encode object primitives independently perform sub-par compared to NSI. The pre-trained DINO backbone is crucial for textured objects and real-world generalization, as evidenced by CLEVrTex/COCO results.

(c) **Non-compositional embeddings inadequately capture object semantics:** CLIP embeddings fall short in capturing object properties beyond basic semantic categories. Moreover, natural language appears insufficiently equipped for object-centric grounding (Chandu et al., 2021).

**Qualitative interpretation:** In Fig. 6(d),(e),(f), we visualize the slot-object pairs inferred by our scoring metric. Interpretable and dense correspondences emerge from NSI contrastive learning. In real-world COCO scenes, we find that slots ground as many as ten objects. See Appendix E.2.6 for more results.

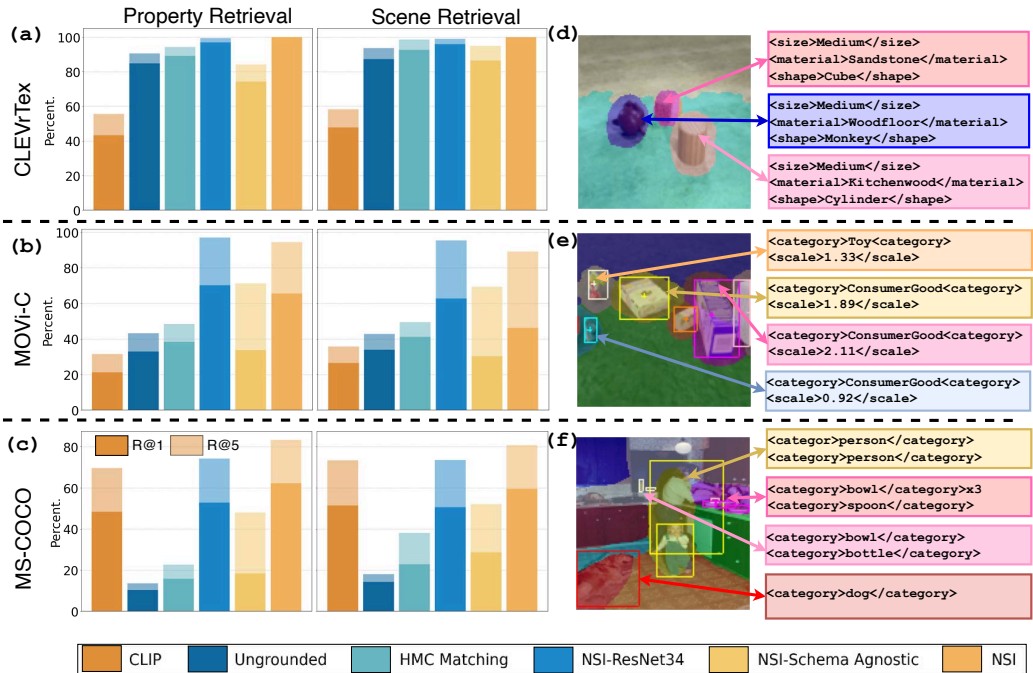

Figure 6: Retrieval results. (a), (b), (c) Property and scene retrieval results. We report Recall@1/5 (higher is better). Standard deviation (over five seeds) was $< 0.3$ across all model instances and retrieval tasks. (d), (e), (f) Visualization of correspondences learned by the NSI similarity metric. The colored arrows show the respective correspondences of schema primitives to the slots. Each schema instance is chunked and color-coded by the slot to which its primitives are assigned.

## 4.4 GROUNDED SLOTS AS VISUAL TOKENS

Grounding-agnostic patch-based tokens are the *de facto* standard for transformer-based models. On the other hand, humans can flexibly abstract out entities free of rigid geometric templates. Here, we investigate the ability of grounded slots to bridge this abstraction gap by training a ViT architecture on slot-based tokenizers for a few-shot classification task. The test suites are derived from the CLEVr-Hans reasoning benchmark (Stammer et al., 2020) that consists of multiple classes based on object attributes and relations. Moreover, the true membership properties are confounded with other attributes in the train split. We explore the following tokenization schemes:

(a) Traditional **patch** tokens extracted from $14 \times 14$ image patches.
(b) **Ungrounded** slot tokens learned from autoencoder object-centric learning on CLEVr.
(c) **Conditional Slot Attention (CoSA)** tokens (Kori et al., 2024) derived from set-matching.
(d) **NSI-CLEVrTex** slot attention trained on semantically-similar CLEVrTex via NSI. The backbone is frozen and subsequently used to infer CLEVr slots, making it *partially grounded*.
(e) *Fully grounded* **NSI-CLEVr** slot attention trained on CLEVr via NSI.

In Fig. 7(a),(b), we report test accuracy against the $k-$shot training sweep for CLEVr-Hans 3 and CLEVr-Hans 7. We found that

(a) **Grounded slots facilitate improved reasoning** and surpass patch-based tokenizers across data regimes using $25\times$ fewer tokens. While the performance of the patch-tokens saturates, grounded slots deconfound object attributes with greater ability with increasing data.
(b) **Partial grounding is often sufficient:** NSI-CLEVrTex slots that are grounded in a different dataset demonstrate competitive reasoning capabilities.
(c) **Property prediction ability does not necessarily transfer to class prediction:** CoSA slots trained on object-attribute prediction show weaker transfer on the Hans-3 dataset.

**How many *visXML* annotations are essential?** In Fig. 7(c), when ablating the number of annotations used to train the NSI-CLEVr tokenizer, we observe that inductive biases instilled by grounding are key, as seen from the sensitivity of the performance to the number of examples. On the other

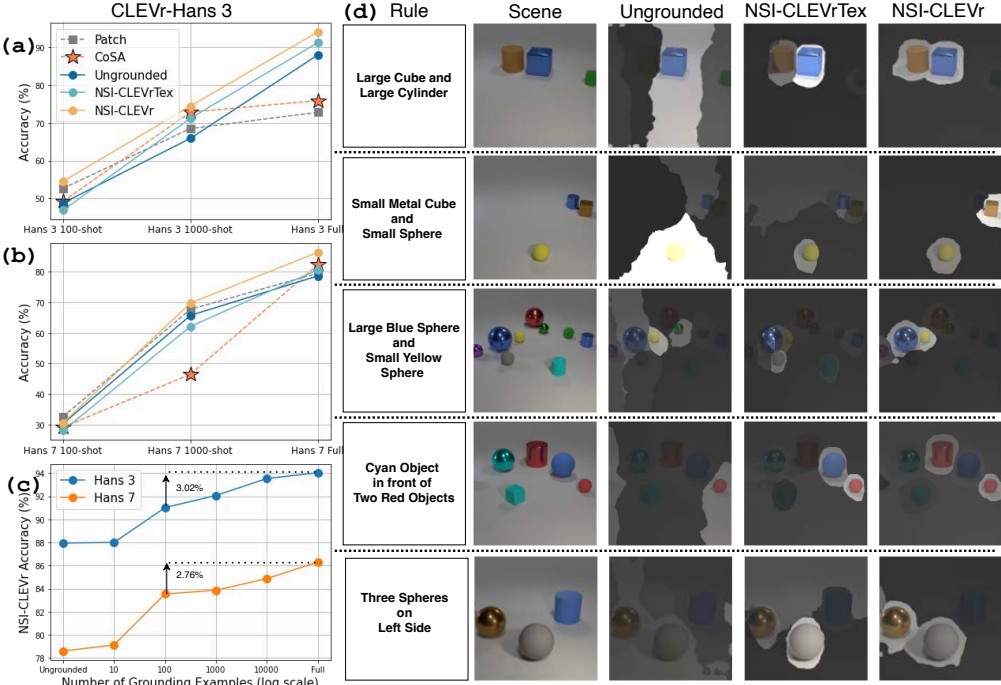

Figure 7: Few-shot classification results. (a) (b) Few-shot classification accuracy on Hans 3 and Hans 7 datasets, respectively. (c) Task accuracy on NSI-CLEVr tokenizers trained on different numbers of grounding examples. Standard deviation (over five seeds) was < 0.03 across tasks and methods. (d) Visual rationales generated across the grounding continuum by extracting attention maps from the final ViT layer. See Appendix E.3.2 for more examples.

hand, significant annotation of scenes is not necessary. Annotating just 100 *visXML* examples yields performative accuracy within a 3% margin of the tokenizer trained on the complete set.

**Qualitative interpretation:** We probe the attention maps from the final layer of the ViT to generate visual rationales. Fig. 7(d) visualizes the maps across the grounding continuum. The ungrounded slots show little to no correlation with the class rule. On the other hand, ViT trained on the NSI-CLEVr slots weighs slots pertinent to the class rule with greater attention. Grounded slots provide abstractions for reasoning that are not only compute-efficient substrates but also yield interpretable visual rationales.

## 5 CONCLUSION

This work introduced NSI, which grounds object semantics into slots for object-centric understanding. It uses a simple schema abstraction to define object concepts and learns to flexibly associate neural embeddings of the schema primitives with object slots via contrastive learning. Unlike set-matching approaches, which struggle when scaled to real-world scenes, NSI enhances object discovery compared to ungrounded counterparts. Further, NSI facilitates interpretable grounding in slot representations. Whereas natural language-grounded embeddings struggle to retrieve granular object properties, we find that NSI embeddings abstracted from the slot-schema intermodal alignment are key representations for such tasks. Finally, we demonstrated the usefulness of NSI as a grounding-aware visual tokenizer that improves the few-shot visual reasoning abilities of ViTs on a hard classification task. While annotations can be prohibitively expensive and laborious, we also demonstrated performative reasoners under practical annotation settings. Slot representations have traditionally been associated with visual stimuli, but object concepts transcend perception to other sensorimotor experiences like audio, tactile signals, and motor behaviors. To this end, NSI lays the groundwork for multimodal object-centric learning. Future work involves adopting NSI to ground common object-centric concepts into different sensorimotor experiences as a step towards a modular human-like understanding of the world.

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

# Appendix

# A PRELIMINARIES

We detail a few essential preliminaries in this section. Readers should refer to the original manuscripts for further details.

## A.1 SLOT ATTENTION

Object-centric learning frameworks decompose scenes by organizing them into semantic object files called slots. Slot Attention (SA) (Locatello et al., 2020) is a powerful iterative attention mechanism for learning such slots from perceptual features extracted from vision backbones. At iteration $t$, for $N$ features $H \in \mathbb{R}^{L \times c}$ and $K$ slots $S^t \in \mathbb{R}^{K \times d}$, the slots compete to explain the features as follows:

$$M = \frac{K(H)Q(S^t)^T}{\sqrt{D}} \in \mathbb{R}^{L \times K}; A_{ij} = \frac{e^{M_{ij}}}{\sum_{j \in \{1, \cdots, K\}} e^{M_{ij}}} \tag{11}$$

$$S^{t+1} = W^T V(S^t) \quad \text{where} \quad W_{ij} = \frac{A_{ij}}{\sum_{i \in \{1, \cdots, N\}} A_{ij}} \tag{12}$$

Here, $Q$, $K$, $V$ are learned query, key, and value matrices, respectively. Note that we present only the attention operations of SA and refer the reader to the original article by (Locatello et al., 2020) to understand the complete rollout.

## A.2 DINO VISION TRANSFORMER

DINO (Caron et al., 2021) is a self-supervised knowledge distillation technique that leads to emergent objectness biases in Vision Transformers (ViTs). These inductive biases make them excellent candidates for learning object representations. A recent object-centric learning method (Seitzer et al., 2023) uses the DINO backbone to scale to complex real-world datasets and reports state-of-the-art results. Motivated by these findings, we use DINO to extract perceptual features for SA.

# B DATA

This section contains additional information on the datasets, *visXML* schema space instantiation, and *visXML* examples.

## B.1 *visXML* SCHEMA SPACE DESCRIPTION

Table 2 lists the various object properties used to create *visXML* description of scenes.

| Dataset | Property | Discrete/Continuous | Size |
|---|---|---|---|
| CLEVr Hans (Johnson et al., 2016; Stammer et al., 2020) | Material | Discrete | 2 |
| | Color | Discrete | 8 |
| | Shape | Discrete | 3 |
| | Size | Discrete | 2 |
| | Object Position | Continuous | 3 |
| CLEVrTex (Karazija et al., 2021) | Texture | Discrete | 60 |
| | Shape | Discrete | 4 |
| | Size | Discrete | 3 |
| | Object Position | Continuous | 3 |
| MOVi-C (Greff et al., 2022) | Object Category | Discrete | 17 |
| | Object Size | Continuous | 1 |
| | Object Position | Continuous | 2 |
| | Bounding Box | Continuous | 4 |
| MS-COCO 2017 (Lin et al., 2015) | Object Category | Discrete | 90 |
| | Bounding Box | Continuous | 4 |

Table 2: *visXML* schema space across various datasets.

## B.2 DATASET SPLITS

The dataset splits used in this work are detailed in Table 3.

| Name | Train Split Size | Validation Split Size | Test Split Size |
|---|---|---|---|
| CLEVr-Hans 3 | 9000 | 2250 | 2250 |
| CLEVr-Hans 7 | 21000 | 5250 | 5250 |
| CLEVrTex | 37500 | 2500 | 10000 |
| MOVi-C | 198635 | 35053 | 6000 |
| MS COCO 2017 | 99676 | 17590 | 4952 |

Table 3: Dataset splits used in experiments.

## B.3 *visXML* EXAMPLES

Figs. 8-11 show *visXML* descriptions of instances from all the datasets.

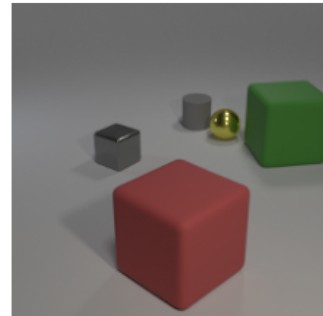

```
<element: 0>
    <size> small</size>
    <shape> cube</shape>
    <color> gray</color>
    <material> metal</material>
    <pos> (-0.76, -0.79, 0.35)</pos>
</element>
<element: 1>
    <size> small</size>
    <shape> sphere</shape>
    <color> yellow</color>
    <material> metal</material>
    <pos> (-0.14, 1.75, 0.35)</pos>
</element>
<element: 2>
    <size> large</size>
    <shape> cube</shape>
    <color> green</color>
    <material> rubber</material>
    <pos> (1.24, 2.12, 0.69)</pos>
</element>
<element: 3>
    <size> small</size>
    <shape> cylinder</shape>
    <color> gray</color>
    <material> rubber</material>
    <pos> (-1.11, 1.79, 0.35)</pos>
</element>
<element: 4>
    <size> large</size>
    <shape> cube</shape>
    <color> red</color>
    <material> rubber</material>
    <pos> (2.73, -2.23, 0.69)</pos>
</element>
```

Figure 8: CLEVr-Hans instance with its corresponding *visXML* description.

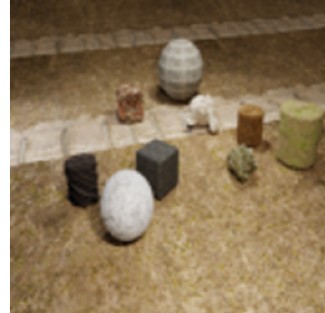

```
<element:0><size>medium</size>
    <shape> sphere</shape>
    <material> whitemarble</material>
    <pos> (1.58, -2.81, 0.60) </pos>
</element>
<element:1><size>small</size>
    <shape> cylinder</shape>
    <material> polyhaven_aerial_mud_1</material>
    <pos> (-0.23, -2.94, 0.40) </pos>
</element>
<element:2><size>medium</size>
    <shape> cylinder</shape>
    <material> polyhaven_forrest_ground_01</material>
    <pos> (2.67, 2.78, 0.60) </pos>
</element>
<element:3><size>medium</size>
    <shape> monkey</shape>
    <material> polyhaven_cracked_concrete_wall</material>
    <pos> (-0.63, 1.98, 0.60) </pos>
</element>
<element:4><size>medium</size>
    <shape> cube</shape>
    <material> polyhaven_brick_wall_005</material>
    <pos> (-2.81, 0.52, 0.42) </pos>
</element>
<element:5><size>large</size>
    <shape> sphere</shape>
    <material> polyhaven_large_grey_tiles</material>
    <pos> (-2.66, 2.94, 0.90) </pos>
</element>
<element:6><size>small</size>
    <shape> cylinder</shape>
    <material> polyhaven_leaves_forest_ground</material>
    <pos> (1.12, 2.49, 0.40) </pos>
</element>
<element:7><size>small</size>
    <shape> monkey</shape>
    <material> polyhaven_aerial_rocks_01</material>
    <pos> (1.98, 0.84, 0.40) </pos>
</element>
<element:8><size>medium</size>
    <shape> cube</shape>
    <material> polyhaven_wood_planks_grey</material>
    <pos> (0.78, -1.23, 0.42) </pos>
</element>
```

Figure 9: CLEVrTex instance with its corresponding *visXML* description.

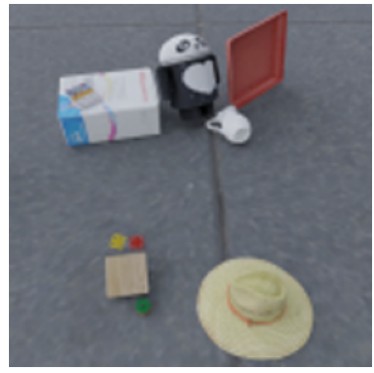

```
<element:0><category>Hat</category>
    <scale> 1.93 </scale>
    <position> (0.68, 0.80)  </position>
    <bbox> (0.70, 0.52, 0.98, 0.84) </bbox>
</element>
<element:1><category>Consumer Goods</category>
    <scale> 2.25 </scale>
    <position> (0.28, 0.28)  </position>
    <bbox> (0.18, 0.14, 0.39, 0.42) </bbox>
</element>
<element:2><category>None</category>
    <scale> 1.98 </scale>
    <position> (0.67, 0.15)  </position>
    <bbox> (0.02, 0.60, 0.29, 0.77) </bbox>
</element>
<element:3><category>Consumer Goods</category>
    <scale> 1.97 </scale>
    <position> (0.50, 0.20)  </position>
    <bbox> (0.09, 0.41, 0.32, 0.58) </bbox>
</element>
<element:4><category>Toys</category>
    <scale> 1.45 </scale>
    <position> (0.33, 0.75)  </position>
    <bbox> (0.63, 0.27, 0.86, 0.39) </bbox>
</element>
<element:5><category>Media Cases</category>
    <scale> 0.80 </scale>
    <position> (0.20, 0.25)  </position>
    <bbox> (0.21, 0.15, 0.29, 0.26) </bbox>
</element>
<element:6><category>None</category>
    <scale> 1.02 </scale>
    <position> (0.62, 0.33)  </position>
    <bbox> (0.28, 0.54, 0.38, 0.66) </bbox>
</element>
```

Figure 10: MOVi-C instance with its corresponding *visXML* description.

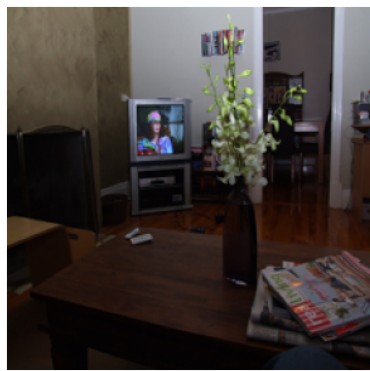

```
<element:0>
    <cat>tv</cat>
    <bbox> (0.34, 0.26, 0.18, 0.17)  </bbox>
</element>
<element:1>
    <cat>chair</cat>
    <bbox> (0.03, 0.32, 0.27, 0.33)  </bbox>
</element>
<element:2>
    <cat>book</cat>
    <bbox> (0.70, 0.67, 0.30, 0.24)  </bbox>
</element>
<element:3>
    <cat>vase</cat>
    <bbox> (0.59, 0.46, 0.10, 0.32)  </bbox>
</element>
<element:4>
    <cat>chair</cat>
    <bbox> (0.72, 0.27, 0.08, 0.22)  </bbox>
</element>
<element:5>
    <cat>dining table</cat>
    <bbox> (0.79, 0.32, 0.07, 0.22)  </bbox>
</element>
<element:6>
    <cat>remote</cat>
    <bbox> (0.34, 0.62, 0.06, 0.04)  </bbox>
</element>
<element:7>
    <cat>book</cat>
    <bbox> (0.71, 0.77, 0.07, 0.09)  </bbox>
</element>
<element:8>
    <cat>chair</cat>
    <bbox> (0.80, 0.28, 0.09, 0.21)  </bbox>
</element>
```

Figure 11: COCO instance with its corresponding *visXML* description.

# C    NEURAL SLOT INTERPRETER

Next, we present details of NSI training modules and summarize the NSI pseudocode.

## C.1    NSI SCENE ENCODER

Fig. 12 shows the DINOSAUR encoder for learning slot representations. Here, the slot attention mechanism operates on pre-trained DINO ViT embeddings. A spatial broadcast MLP reconstructs the feature embeddings from the slots.

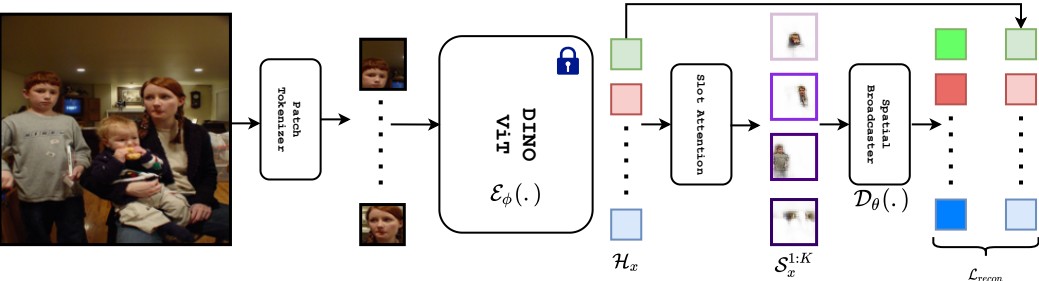

Figure 12: A DINOSAUR encoder (Seitzer et al., 2023) learns to represent images via slots.

## C.2    NSI SCHEMA ENCODER

The NSI schema encoder architecture is shown in Fig. 13. The lower-level primitive encoder represents individual object attributes via primitives and their property tags. However, objects in scenes also exist in their relation to other objects. To this end, a higher-level Schema Transformer represents the primitives jointly.

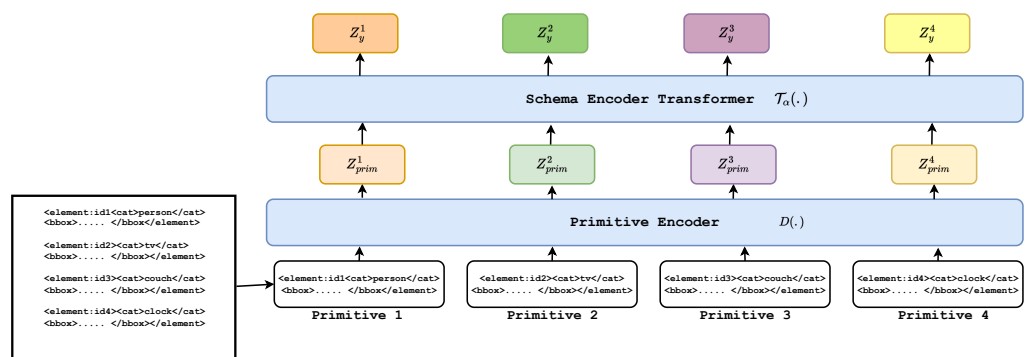

Figure 13: A bi-level schema encoder learns a representation of schema primitives. The primitive encoder embeds the object properties of each schema primitive. Then, a Transformer learns embeddings that assimilate the entire schema context.

## C.3    NSI PSEUDOCODE

Algorithm 1 contains Python-style pseudocode for the NSI metric and subsequent contrastive learning objective.

# D    HYPERPARAMETERS

We list the hyperparameters for NSI and other methods used in our experiments, which were all performed on Nvidia A100 GPUs.

---

**Algorithm 1** Neural Slot Interpreter Contrastive Learning Pseudocode

---

1: **Require:** Projection heads $\mathcal{H}_{scene}(.), \mathcal{H}_{schema}(.)$
2: **Require:** Batch of slot embeddings from the slot encoder $\{S_x^{1:K}\}_{1:B}$
3: **Require:** Batch of primitive embeddings from the schema encoder $\{Z_y^{1:N}\}_{1:B}$
4: # Compositional Score Aggregation
5: $\left\{Y_x^{1:K}\right\}_{1:B} = \mathcal{H}_{scene}\left(\{S_x^{1:K}\}_{1:B}\right)$                    ▷ Project slots
6: $\{Y_y^{1:N}\}_{1:B} = \mathcal{H}_{schema}\left(\{Z_y^{1:N}\}_{1:B}\right)$                    ▷ Project primitives
7: **for all** $i, j \in \{1, \cdots, B\}$ **do**                    ▷ Compute $\mathcal{S} \in \mathbb{R}^{B \times B}$
8:     $\mathcal{S}_{ij} = \sum_{n \in \{1, \cdots, N\}} \max_{k \in \{1, \cdots, K\}} {Y_i^k}^T Y_j^n$
9: **end for**
10: # Contrastive Learning
11: $labels = arange(B)$
12: $\mathcal{L}_{schema} = CrossEntropyLoss\left(labels, \ \mathcal{S}\right)$
13: $\mathcal{L}_{scene} = CrossEntropyLoss\left(labels, \ \mathcal{S}^T\right)$
14: $\mathcal{L}_{contrastive} = (\mathcal{L}_{schema} + \mathcal{L}_{scene})/2$
15: **return** $\mathcal{L}_{contrastive}$

---

### D.1 UNGROUNDED AND HMC MATCHING BACKBONE HYPERPARAMETERS

The hyperparameters for ungrounded and HMC matching backbones are given in Table 4.

### D.2 NSI HYPERPARAMETERS

The hyperparameters for the NSI alignment model are listed in Table 5. The ablated architectures follow the same setup without the ablated module.

| Module | Hyperparameters | CLEVr-Hans | CLEVrTex | MOVi-C | MS COCO 2017 |
|---|---|---|---|---|---|
| DINO Backbone | Image Size | 224 | 224 | 224 | 224 |
| | Patch Size | 8 | 8 | 8 | 8 |
| | Num. Patches | 784 | 784 | 784 | 784 |
| | Num. Layers | 8 | 8 | 8 | 8 |
| | Num. Heads | 8 | 8 | 8 | 8 |
| | Hidden Dims. | 192 | 192 | 192 | 192 |
| Slot Attention | Num. Slots | 10 | 10 | 10 | 93 |
| | Iterations | 3 | 3 | 3 | 3 |
| | Hidden Dims. | 192 | 192 | 192 | 192 |
| Broadcast Decoder | Num. MLP Layers | 3 | 3 | 3 | 3 |
| | MLP Hidden Dims. | 1024 | 1024 | 1024 | 1024 |
| | Output Dims. | 785 | 785 | 785 | 785 |
| Prediction Head | MLP Hidden Layers | 2 | 2 | 2 | 2 |
| | MLP Hidden Dims. | 64 | 64 | 64 | 64 |
| | Output Size | 18 | 71 | 25 | 95 |
| Training Setup | Batch Size | 128 | 128 | 128 | 128 |
| | LR Warmup steps | 10000 | 10000 | 10000 | 30000 |
| | Peak LR | $4 \times 10^{-4}$ | $4 \times 10^{-4}$ | $4 \times 10^{-4}$ | $1 \times 10^{-4}$ |
| | Dropout | 0.1 | 0.1 | 0.1 | 0.1 |
| | Gradient Clipping | 1.0 | 1.0 | 1.0 | 1.0 |
| Inference Configuration | Num. Slots | 10 | 10 | 10 | 30 |
| Training Cost | GPU Usage | 40 GB | 40 GB | 40 GB | 40 GB |
| | Days | 1 | 3 | 3 | 5 |
| ViT | Num Layers | 2 | - | - | - |
| | Hidden Dims | 64 | - | - | - |
| | Num Heads | 4 | - | - | - |

Table 4: Hyperparameters for the ungrounded and HMC matching method used in our experiments. In the ungrounded case, the backbone and slot attention modules are trained solely on the reconstruction objective and frozen.

| Module | Hyperparameters | CLEVr Hans | CLEVrTex | MOVi-C | MS COCO 2017 |
|---|---|---|---|---|---|
| DINO Backbone | Image Size | 224 | 224 | 224 | 224 |
| | Patch Size | 8 | 8 | 8 | 8 |
| | Num. Patches | 784 | 784 | 784 | 784 |
| | Num. Layers | 8 | 8 | 8 | 8 |
| | Num. Heads | 8 | 8 | 8 | 8 |
| | Hidden Dims. | 192 | 192 | 192 | 192 |
| Broadcast Decoder | Num. MLP Layers | 3 | 3 | 3 | 3 |
| | MLP Hidden Dims. | 1024 | 1024 | 1024 | 1024 |
| | Output Dims. | 785 | 785 | 785 | 785 |
| Slot Attention | Num. Slots | 10 | 10 | 10 | 15 |
| | Iterations | 3 | 3 | 3 | 3 |
| | Hidden Dims. | 192 | 192 | 192 | 192 |
| Schema Encoder | Num. Layers | 8 | 8 | 8 | 8 |
| | Num. Heads | 8 | 8 | 8 | 8 |
| | Hidden Dims. | 192 | 192 | 192 | 192 |
| | Max. Schema Len. | 10 | 10 | 10 | 93 |
| Projection Heads | Embedding Dims. | 64 | 64 | 64 | 64 |
| | MLP Hidden Layers | 2 | 2 | 2 | 2 |
| | MLP Hidden Dims. | 256 | 256 | 256 | 256 |
| Training Setup | Batch Size | 128 | 128 | 128 | 128 |
| | LR Warmup steps | 10000 | 10000 | 10000 | 30000 |
| | Peak LR | $4 \times 10^{-4}$ | $4 \times 10^{-4}$ | $4 \times 10^{-4}$ | $1 \times 10^{-4}$ |
| | Dropout | 0.1 | 0.1 | 0.1 | 0.1 |
| | Gradient Clipping | 1.0 | 1.0 | 1.0 | 1.0 |
| | $\beta_1, \beta_2$ | 0.5, 0.5 | 0.5, 0.5 | 0.5, 0.5 | 0.5, 0.5 |
| Training Cost | GPU Usage | 40GB | 40 GB | 40 GB | 40 GB |
| | Days | 1 | 3 | 3 | 5 |
| ViT | Num Layers | 2 | - | - | - |
| | Hidden Dims | 64 | - | - | - |
| | Num Heads | 4 | - | - | - |

Table 5: Hyperparameters for the NSI alignment model instantiation and training setup.

# E EXPERIMENTS

## E.1 OBJECT DISCOVERY

### E.1.1 OBJECT DISCOVERY RESULTS

| Dataset | Metric | Ungrounded | HMC Matching | NSI |
|---|---|---|---|---|
| CLEVrTex | FG-ARI | $87.79 \pm 0.12$ | $88.37 \pm 0.12$ | $89.89 \pm 0.01$ |
| | mBO | $44.86 \pm 0.04$ | $45.23 \pm 0.23$ | $46.60 \pm 0.02$ |
| MOVi-C | FG-ARI | $65.53 \pm 0.15$ | $65.61 \pm 0.31$ | $66.41 \pm 0.12$ |
| | mBO | $36.79 \pm 0.03$ | $36.79 \pm 0.41$ | $38.52 \pm 0.23$ |
| COCO | FG-ARI | $40.12 \pm 0.29$ | $32.18 \pm 0.45$ | $44.24 \pm 0.27$ |
| | mBO$^i$ | $27.20 \pm 0.31$ | $18.32 \pm 0.51$ | $28.12 \pm 0.25$ |
| | mBO$^c$ | $26.54 \pm 0.25$ | $19.61 \pm 0.82$ | $32.10 \pm 0.31$ |
| PASCAL VOC 2012 (Zero-Shot) | FG-ARI | $20.42 \pm 0.13$ | $15.14 \pm 0.09$ | $21.97 \pm 0.17$ |
| | mBO$^i$ | $35.97 \pm 0.15$ | $25.15 \pm 0.08$ | $36.98 \pm 0.19$ |
| | mBO$^c$ | $37.94 \pm 0.17$ | $27.02 \pm 0.12$ | $39.06 \pm 0.22$ |

Table 6: Object Discovery results. We use the DINO backbone and an MLP decoder across methods. The standard deviation was calculated over five random seeds.

Table 6 contains the complete set of object discovery results with the standard deviations. We also conducted a zero-shot evaluation of the models trained on MS-COCO on Pascal VOC 2012 (Everingham et al., 2015). We observed that the benefits of grounding the model via NSI on one real-world dataset were transferred to the other for object discovery, with the grounded model also improving mask segmentation over its ungrounded counterparts for Pascal VOC.

### E.1.2 SLOT SWEEP OVER COCO SCENES

Fig. 14 shows the NSI object discovery results on COCO scenes as the number of slots is increased. We observe that segmentation improves until 15 slots and then slowly tapers off.

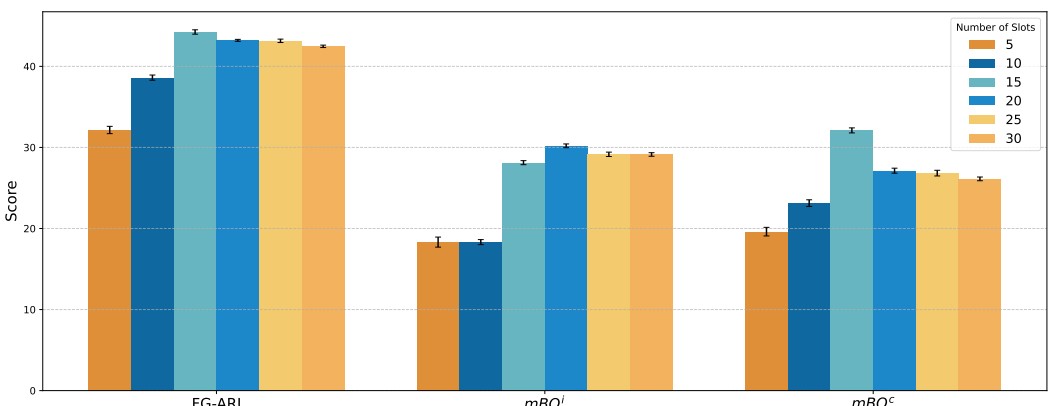

Figure 14: Ablation on the number of slots for COCO object discovery using NSI.

### E.1.3 PROPERTY ABLATIONS ON COCO SCENES

In Fig. 15, we ablate the properties used to form schema primitives. Ostensibly, the slots weigh bounding box coordinates more than categories, as the performance drop is steeper when the former is ablated as opposed to the meagre loss when the latter is ablated.

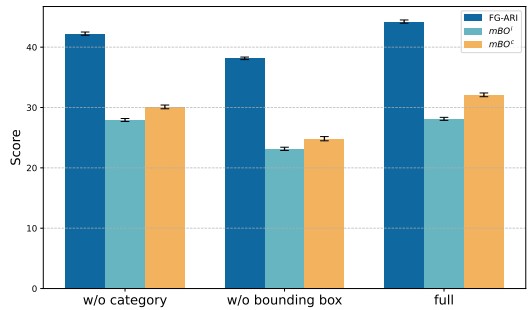

Figure 15: Ablation on the *visXML* properties for NSI.

### E.1.4 GROUNDING EXAMPLES ABLATION ON COCO SCENES

Figure 16 shows the results of ablating over the number of grounding examples used for NSI co-training. We observe that the $ARI$ metric and the class-wise $mBO$ are sensitive to grounding, with as few as 100 annotations improving object discovery via segmentation masks.

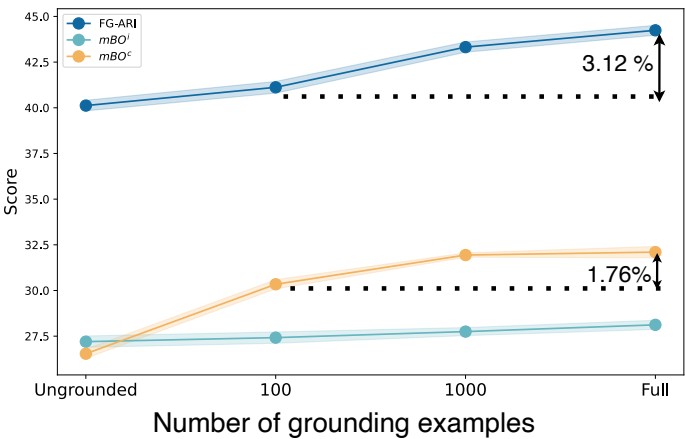

Figure 16: Ablation on number of grounding examples used to train NSI on MS COCO.

### E.1.5 MASK VISUALIZATIONS

Figs. 17, 18, 19, 20 visualize slot masks over scenes from CLEVrTex, MOVi-C, COCO, and Pascal VOC, respectively.

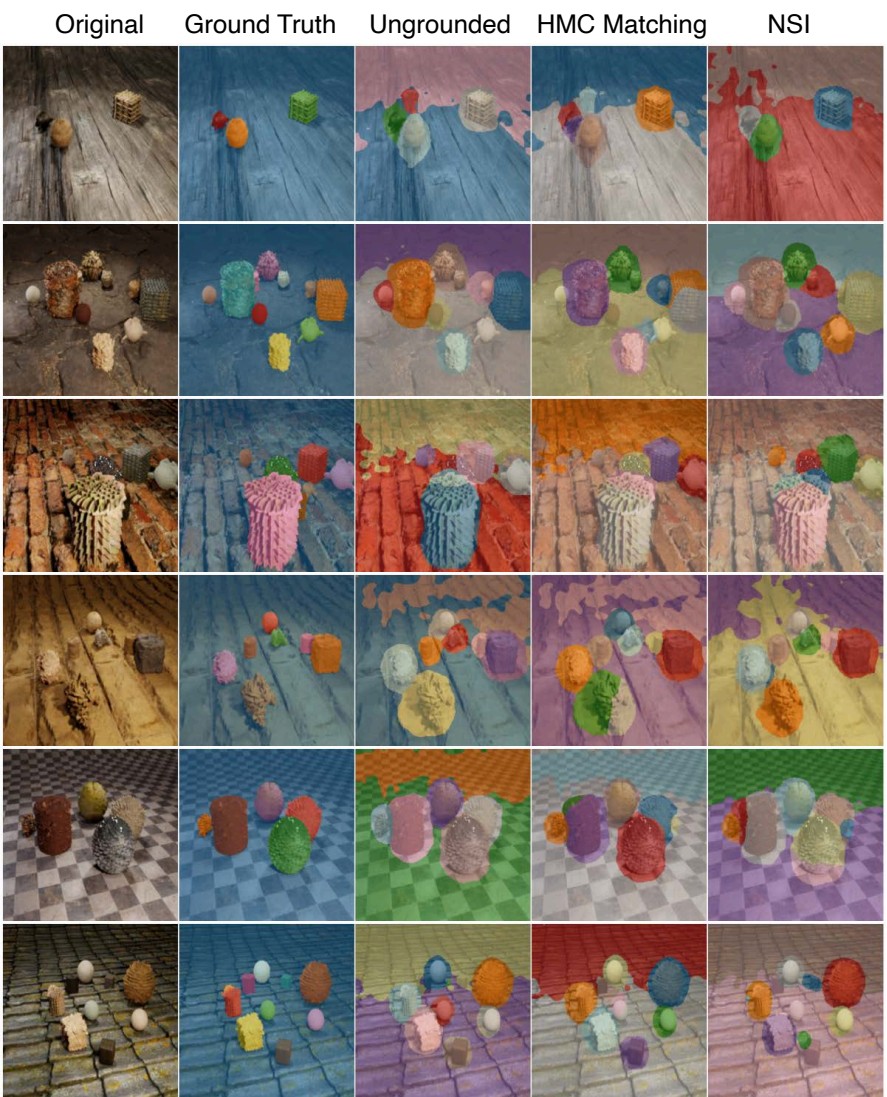

Figure 17: Object discovery results on CLEVrTex scenes.

Original  Ground Truth  Ungrounded  HMC Matching  NSI

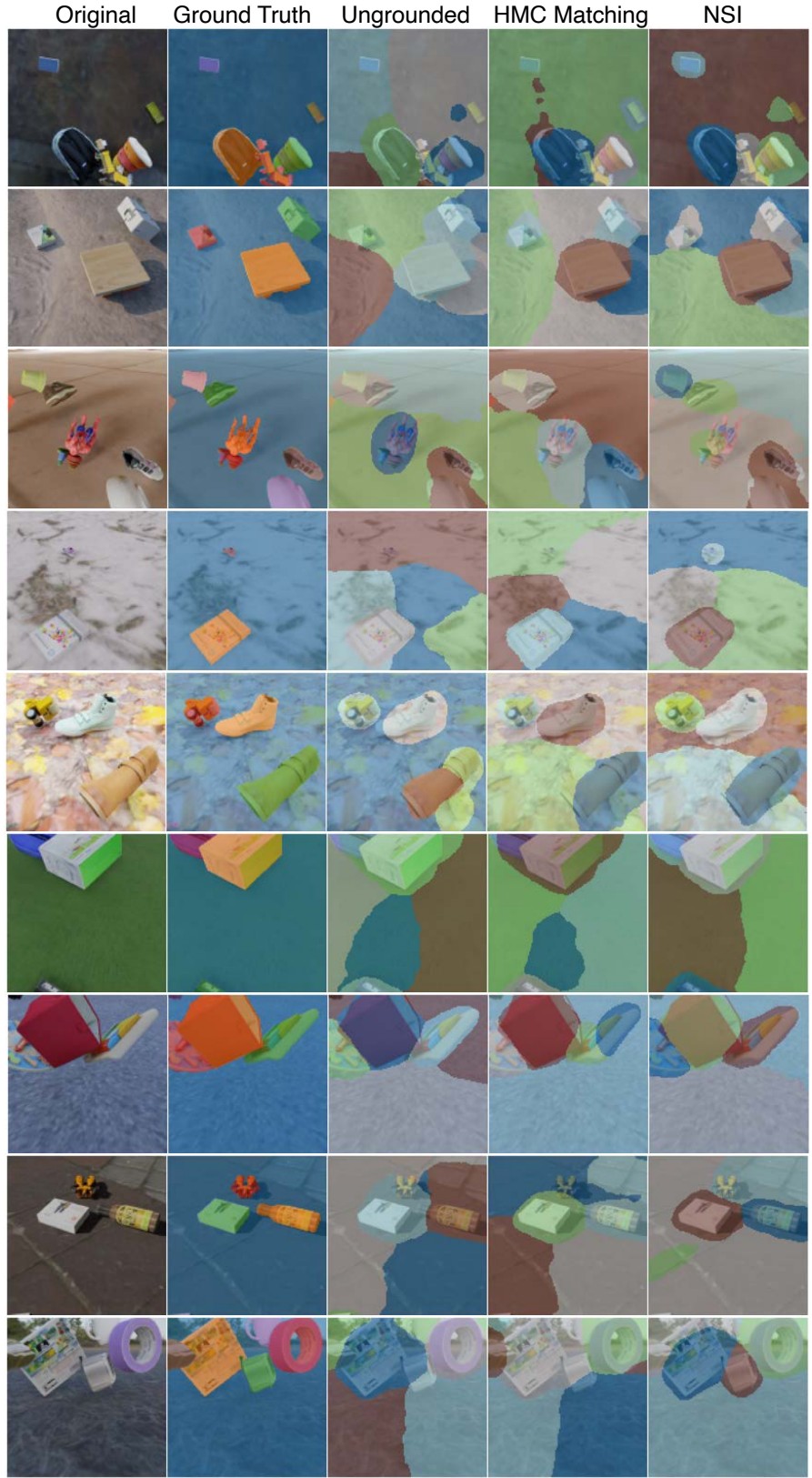

Figure 18: Object discovery results on MOVi-C scenes.

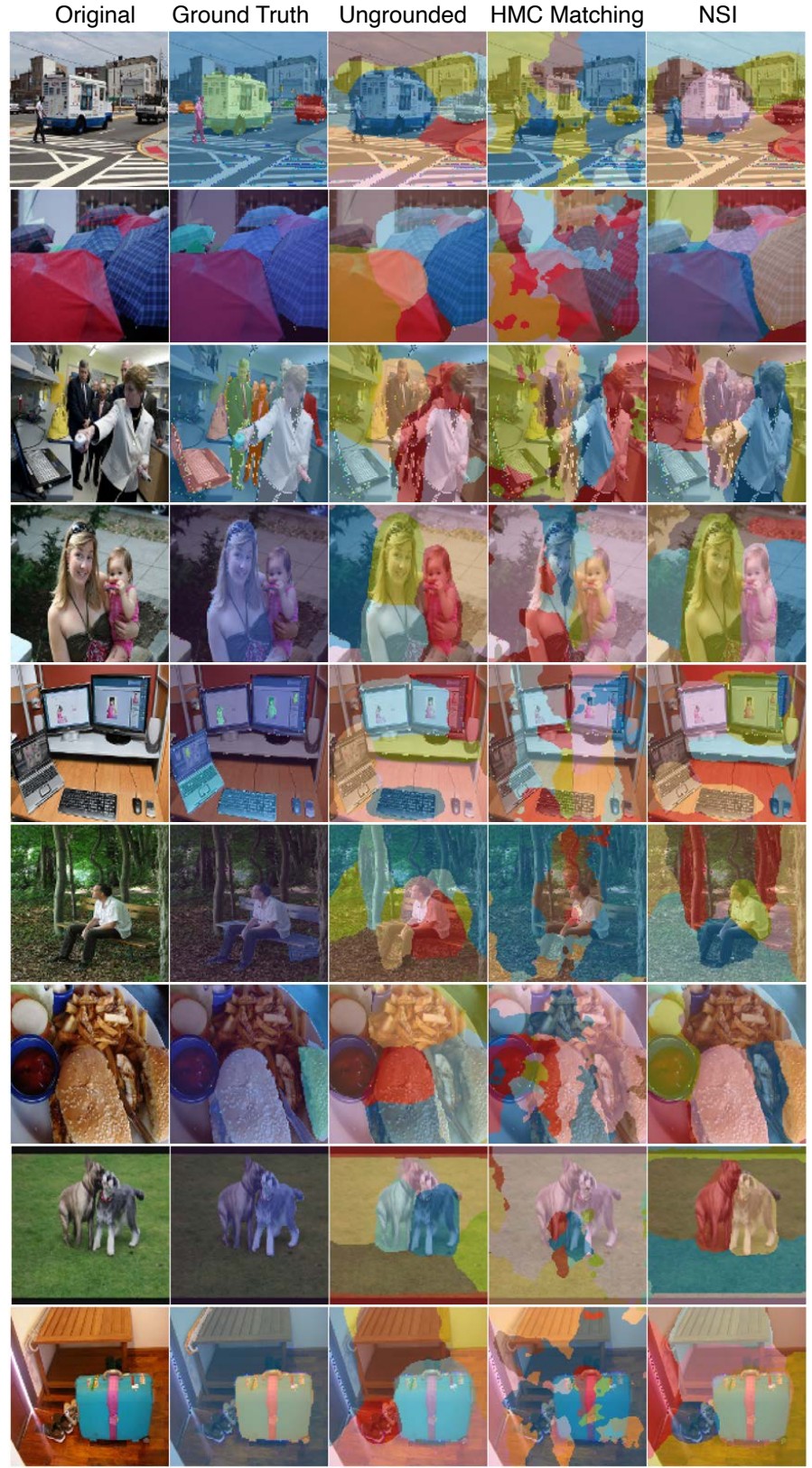

Figure 19: Object discovery results on COCO scenes.

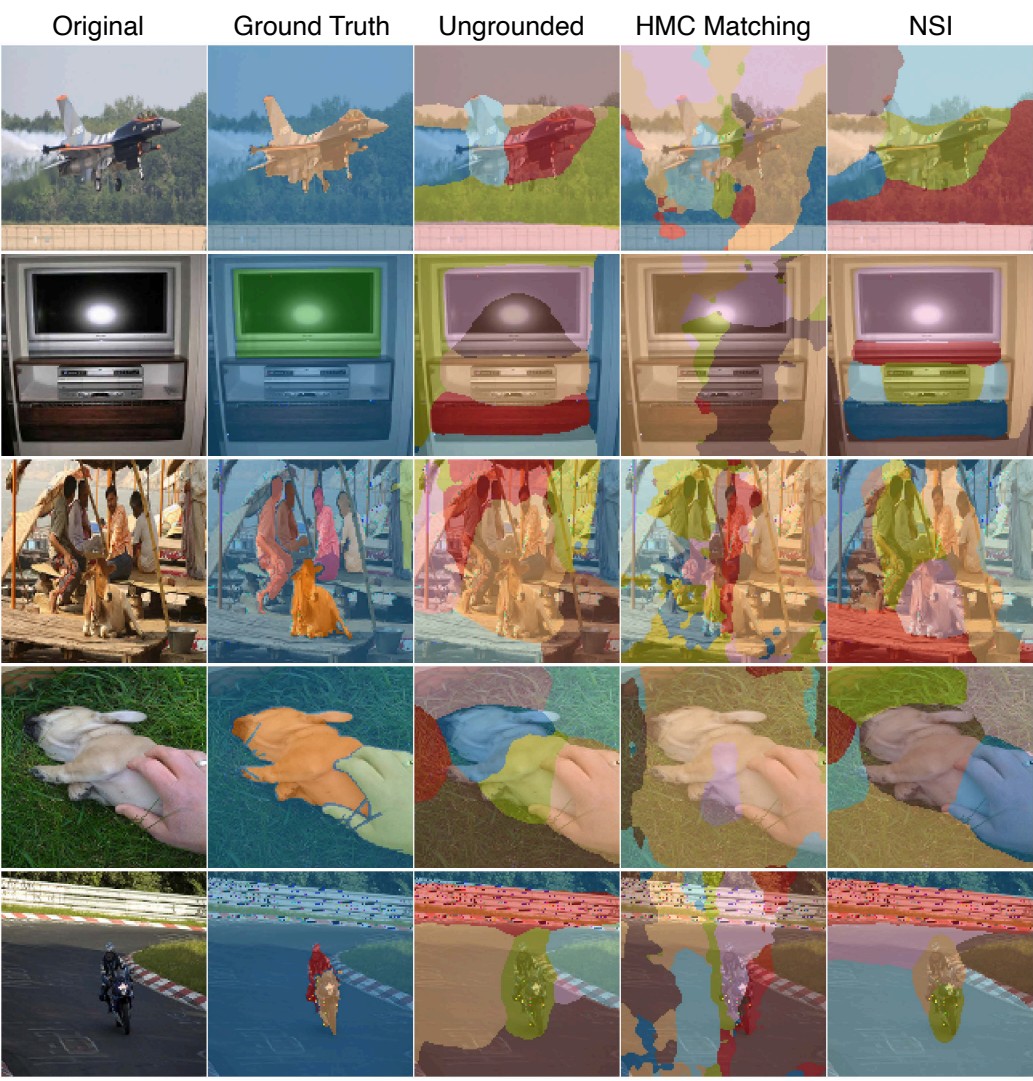

Figure 20: Zero-shot object discovery results on Pascal VOC 2012 scenes.

## E.2 GROUNDED COMPOSITIONAL SEMANTICS

### E.2.1 TOP LEARNED SLOTS

Figs. 21-23 show the top 50 slots learned by the image encoder over each dataset. Each slot is weighted by the $l_2$ norm of its embeddings $Y_x^{rk}$. The top slots follow an interesting distribution. On CLEVrTex (Fig. 21), slots with larger objects and a clean segmentation are assigned a higher magnitude. Intuitively, these slots are the most discriminative when assigning primitives that contain shape, texture, and size, and benefit from large, cleanly segmented objects. On the other hand, the emergent distribution in MOVi-C (Fig. 22) and COCO (Fig. 23) weights edge artifacts more. We posit that since the object annotations of these datasets contain object positions in the image, edge objects tend to be more discriminative.

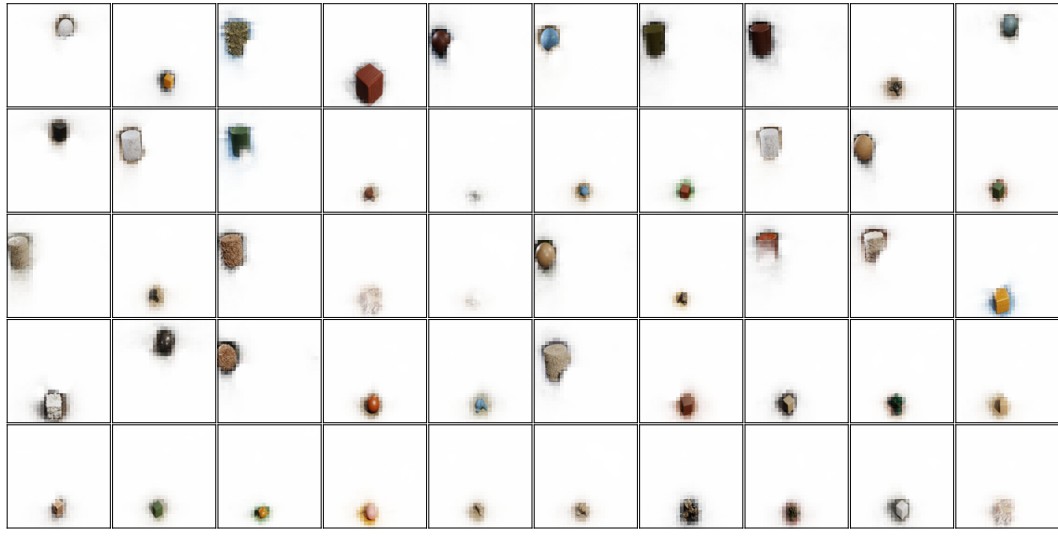

Figure 21: Top 50 slots (left to right, top to bottom) ranked by magnitude from test split of the CLEVrTex dataset.

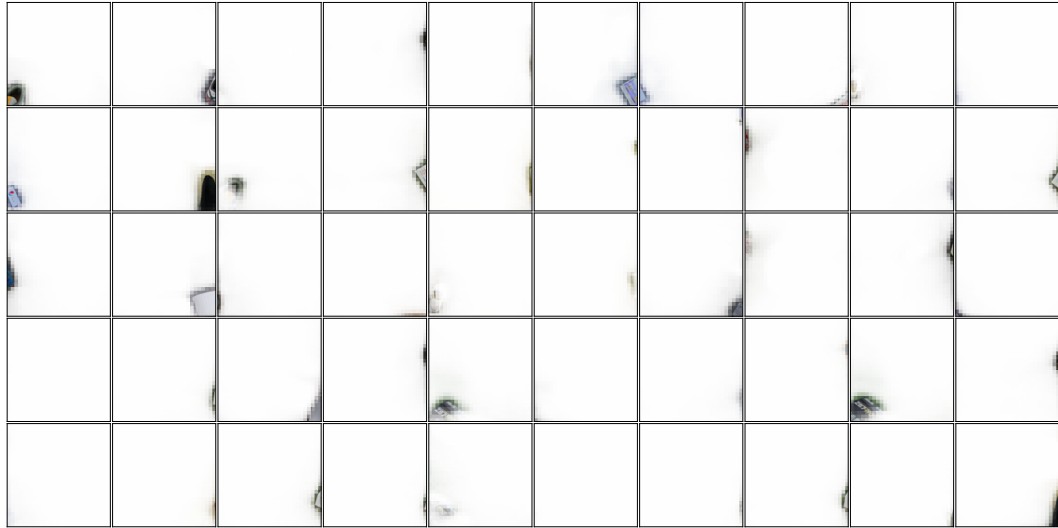

Figure 22: Top 50 slots (left to right, top to bottom) ranked by magnitude from test split of the MOVi-C dataset.

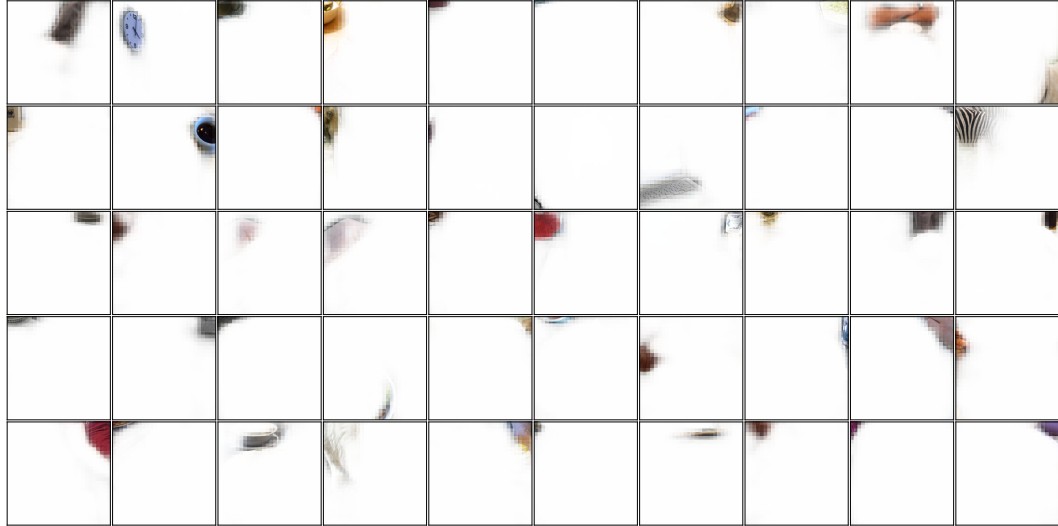

Figure 23: Top 50 slots (left to right, top to bottom) ranked by magnitude from test split of the COCO dataset.

### E.2.2 TOP LEARNED PRIMITIVES

Figs. 24-26 show the t-SNE visualization (van der Maaten & Hinton, 2008) of representations learned by the schema encoder on the *visXML* primitives. In each plot, primitives in the test split are encoded context-free. The learned embeddings exhibit clustering effects on property categories.

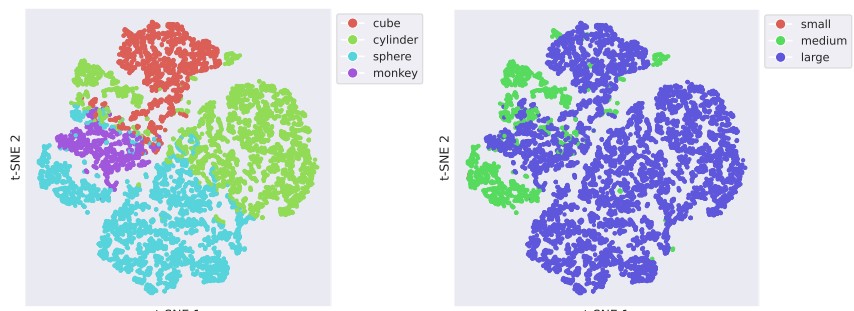

(a) t-SNE scatter plot labeled with object shapes.

(b) t-SNE scatter plot labeled with object sizes.

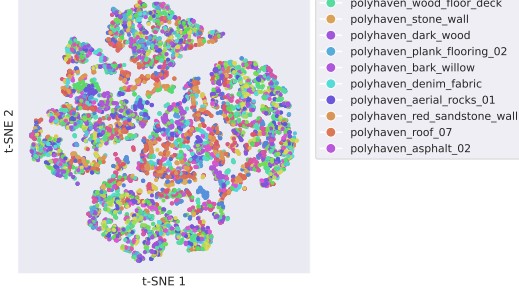

(c) t-SNE scatter plot labeled with object materials.

Figure 24: t-SNE scatter plots of the top 10000 CLEVrTex primitives weighted by the $l_2$ norm of their embeddings. The embeddings are clearly clustered by the shape type. Interestingly, when looking at the object size, only large and medium-sized objects are represented in the top primitives.

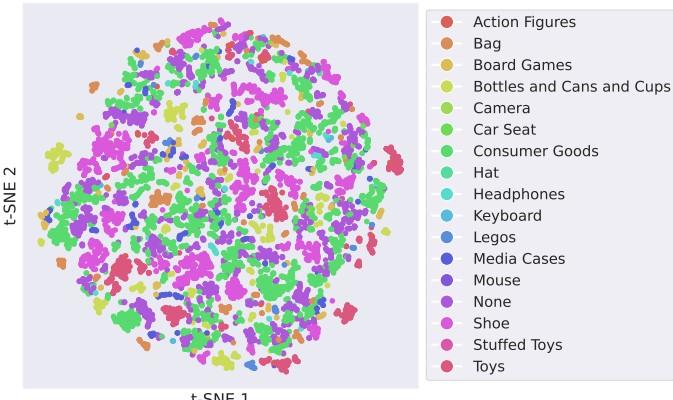

Figure 25: t-SNE scatter plots of the top 10000 MOVi-C primitives weighted by the $l_2$ norm of their embeddings. The scatter points are labeled with the object categories. While there are multiple local clusters, a larger clustering effect based on object categories are not apparent.

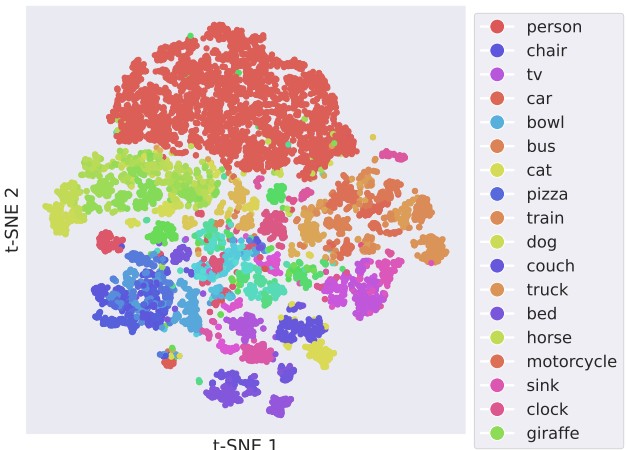

Figure 26: t-SNE scatter plots of the top 10000 COCO primitives weighted by the $l_2$ norm of their embeddings. The scatter points are labeled with the object categories. A category-based clustering pattern is emergent.

### E.2.3 SEARCHING OVER SLOTS

We found that the NSI metric, which learns to match entire schemas to entire images, can also be used zero-shot to reliably retrieve individual slots from primitives. We demonstrate two instances of slot search in Figs. 27 and 28. A query in the form of a single primitive is embedded using the schema encoder. The query embedding ranks the slot of a database formed from the test split. In Figs. 27 and 28, we show the property search and position search results, respectively.

### E.2.4 SLOT SWEEP FOR COCO RETRIEVAL TASK

Fig. 29 shows the effect of the number of slots on the recall rates. Recall is highest at 15-20 slots for COCO scenes and the model overfits as the number of slots increases further.

### E.2.5 RETRIEVAL WITH UNDERSPECIFIED ANNOTATIONS

We investigated the efficacy of grounding under the setting where the annotations of scenes were underspecified. Figs. 30(a) and (b) show the retrieval results under two settings: (a) a maximum of ten objects annotated in a scene and (b) a maximum of five objects annotated in a scene. First,

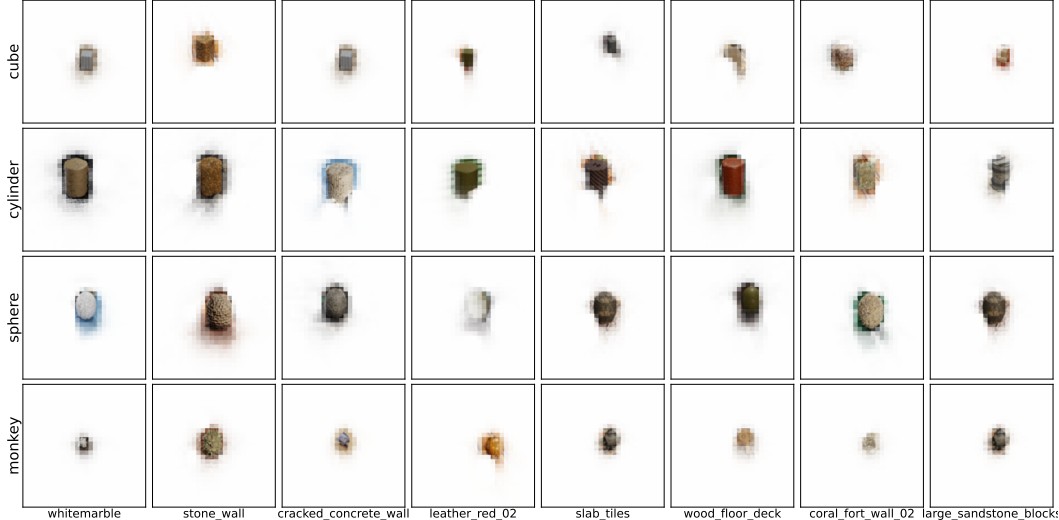

Figure 27: Searching over slots with property queries. The rows represent a shape and the columns represent an object material. The position is fixed at $[0, 0, 0]$ and the size is set to 'Large.' The alignment model reliably retrieves slots with desired object properties if they exist in the database.

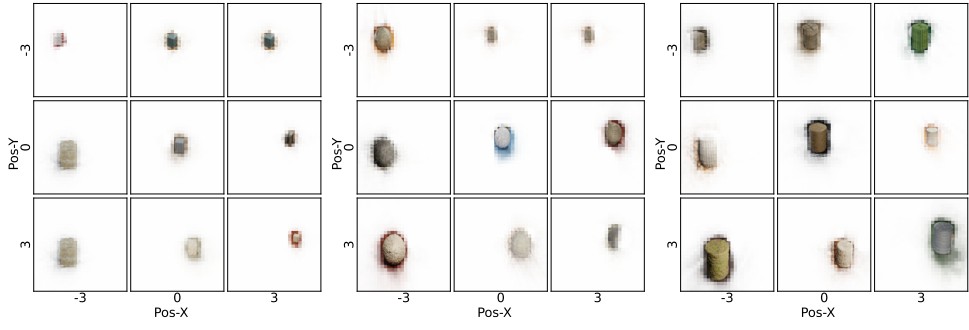

(a) Position search on cubes.    (b) Position search on spheres.    (c) Position search on cylinders.

Figure 28: Searching over slots with position queries. We adjust the 'Pos-X' for each shape on the $x$-axis and 'Pos-Y' on the $y$-axis. The top-ranked slot clearly reflects the position adjustment.

we observed that the drop in recall rates for the ResNet backbone was significant compared to the full setting. Second, CLIP and NSI were resilient to the limited annotation, with NSI outperforming other methods. CLIP benefited from the vast training corpora that helped it generalize to the under-specification. On the other hand, the compositional grounding with strong backbones endows NSI with string retrieval despite being trained on limited data.

### E.2.6 ADDITIONAL QUALITATIVE RESULTS

Fig. 31 shows additional results on the associations inferred by NSI on MOVi-C and MS COCO 2017.

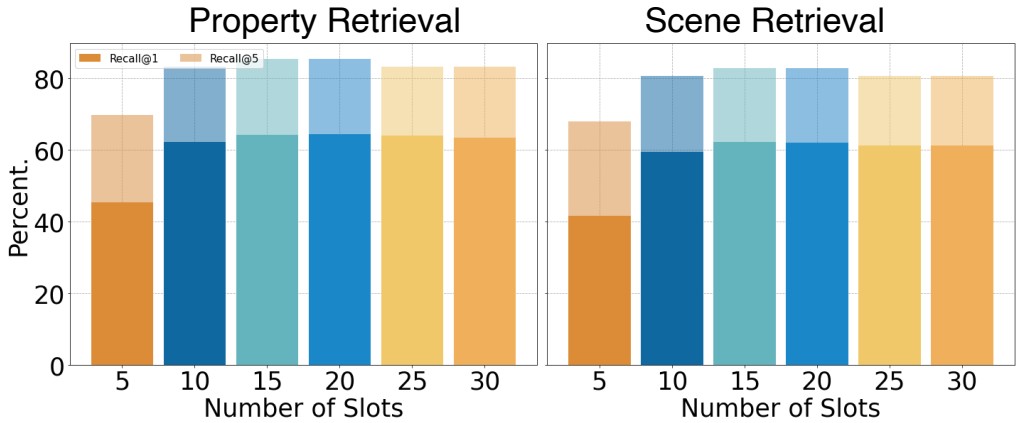

Figure 29: Ablation on the number of slots for scene-property retrieval task. The standard deviation over five seeds was $< 0.3$ across datasets.

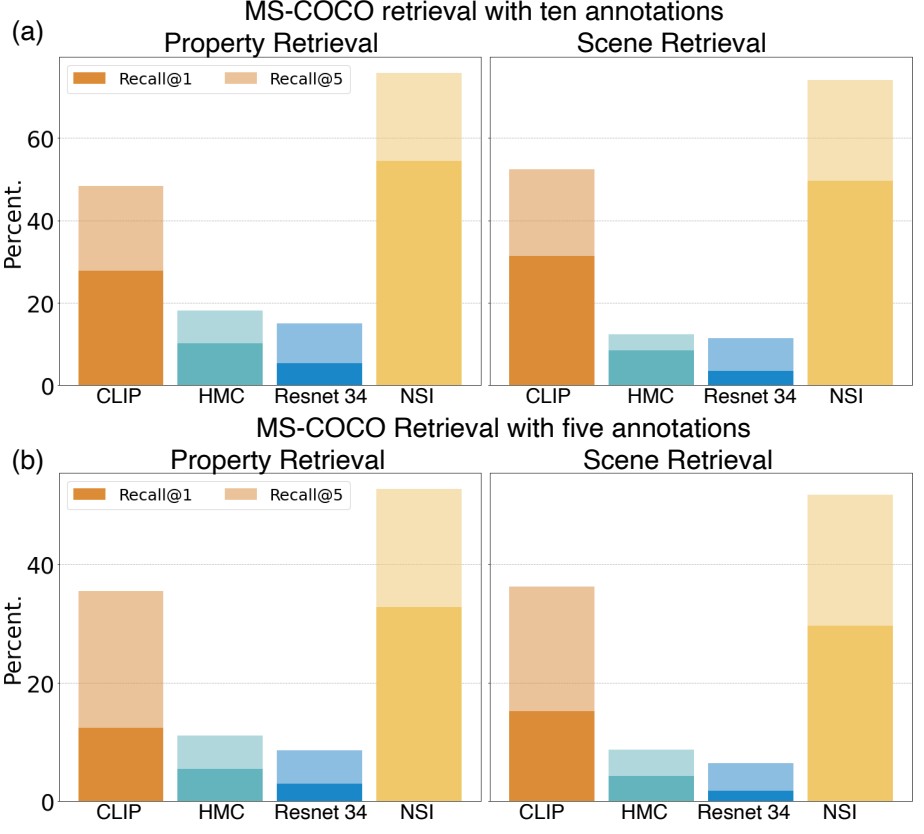

Figure 30: Retrieval results on underspecified scene annotations settings.

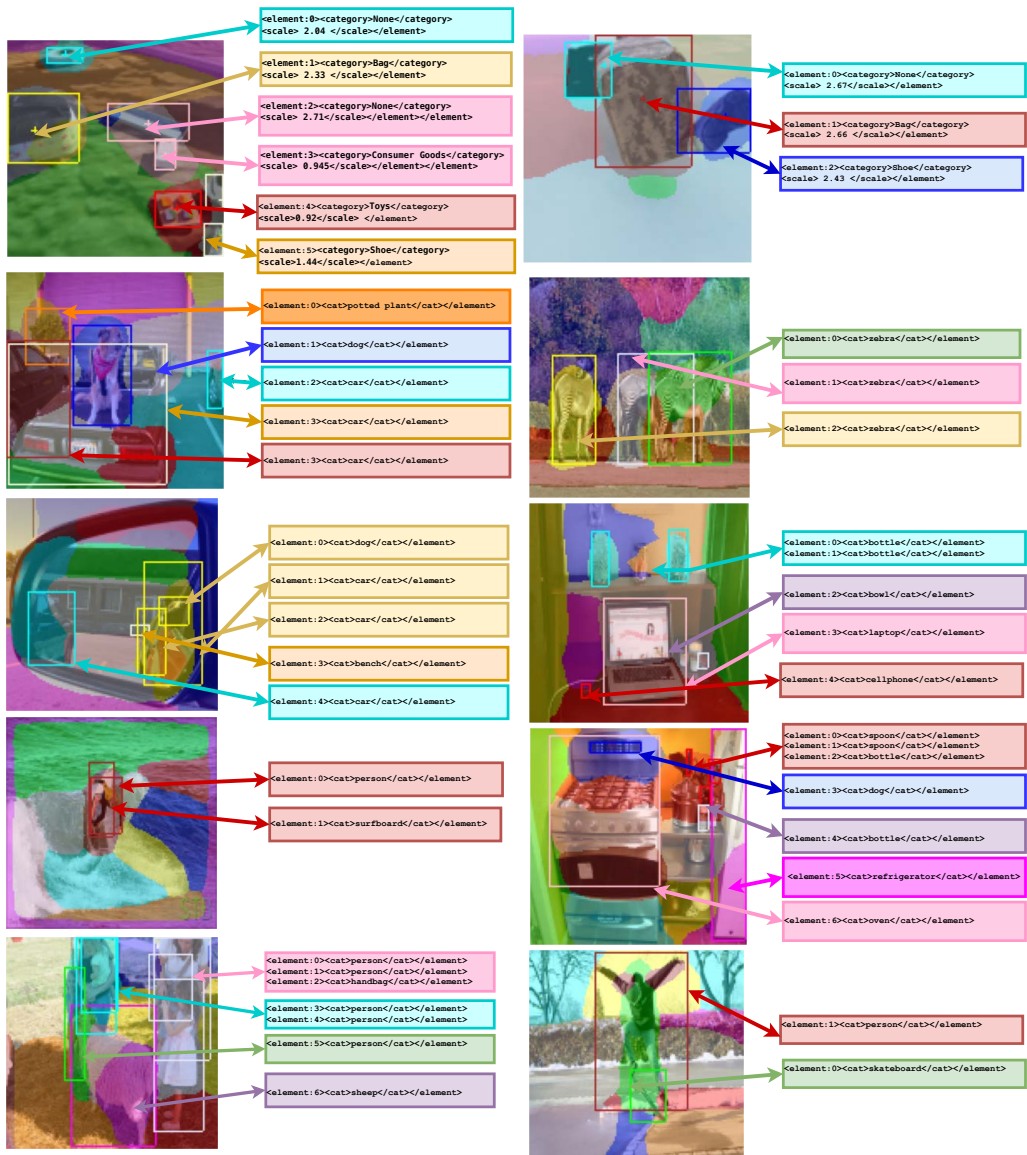

Figure 31: Correspondences inferred by NSI on MOVi-C and COCO scenes.

### E.3 GROUNDED SLOTS AS VISUAL TOKENS

#### E.3.1 CONFUSION MATRICES

Figs. 32 and 33 show the classification confusion matrices for the CLEVr-Hans 3 and CLEVr-Hans 7 tasks. In the few-shot setting, the model often predicts scenes into under-specified and less discriminative classes, like "cyan object in front of two red objects." Similarly, it also tends to mispredict into classes containing large objects like "large cube and large cylinder" that contain reasoning over larger objects and are easier to tokenize.

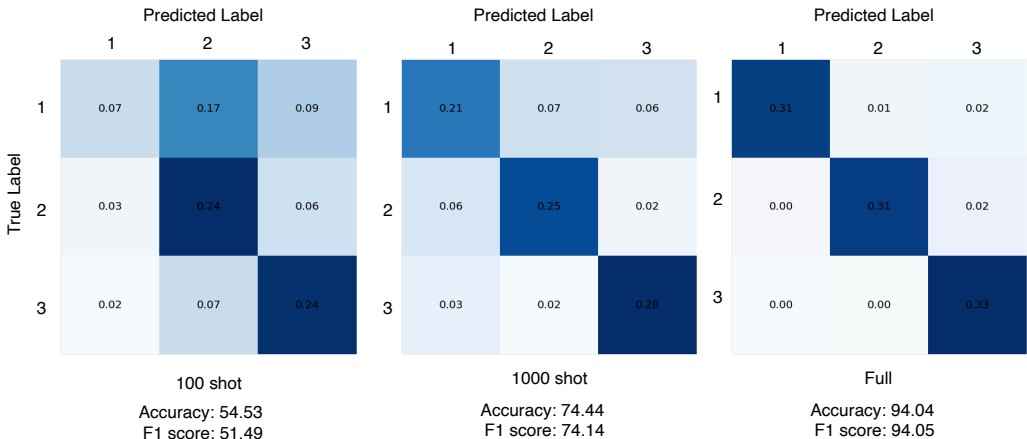

Figure 32: Confusion matrices for NSI prediction on the CLEVr-Hans 3 task.

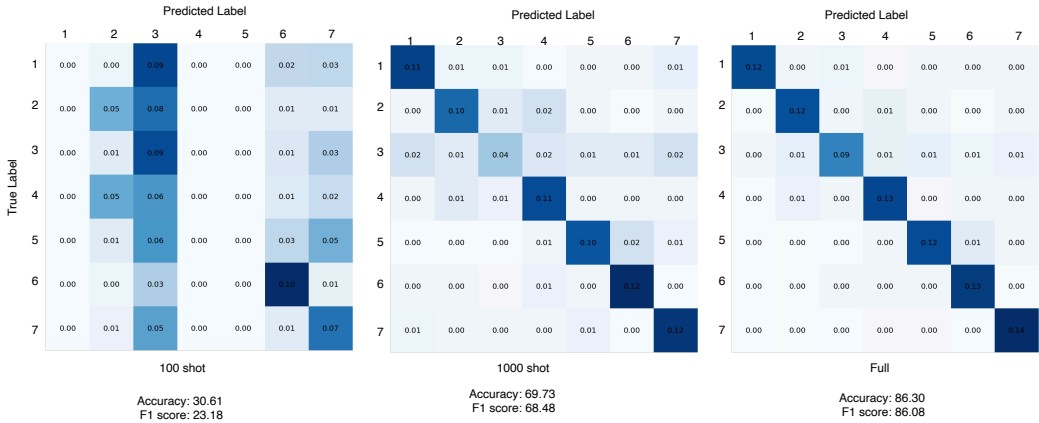

Figure 33: Confusion matrices for NSI prediction on the CLEVr-Hans 7 task.

#### E.3.2 VISUAL RATIONALES

Fig. 34 simultaneously visualizes the rationales across data settings and attention layers. On an average, we observe that rationales tend to get stronger and more accurate as the number of training examples increases and the ViT depth increases.

Figs. 35 and 36 demonstrate rationales across the grounding continuum for Hans 3 and Hans 7 classification tasks, respectively.

Fig. 37 demonstrates attention maps where the model prediction is correct, but the rationale is not entirely dispositive.

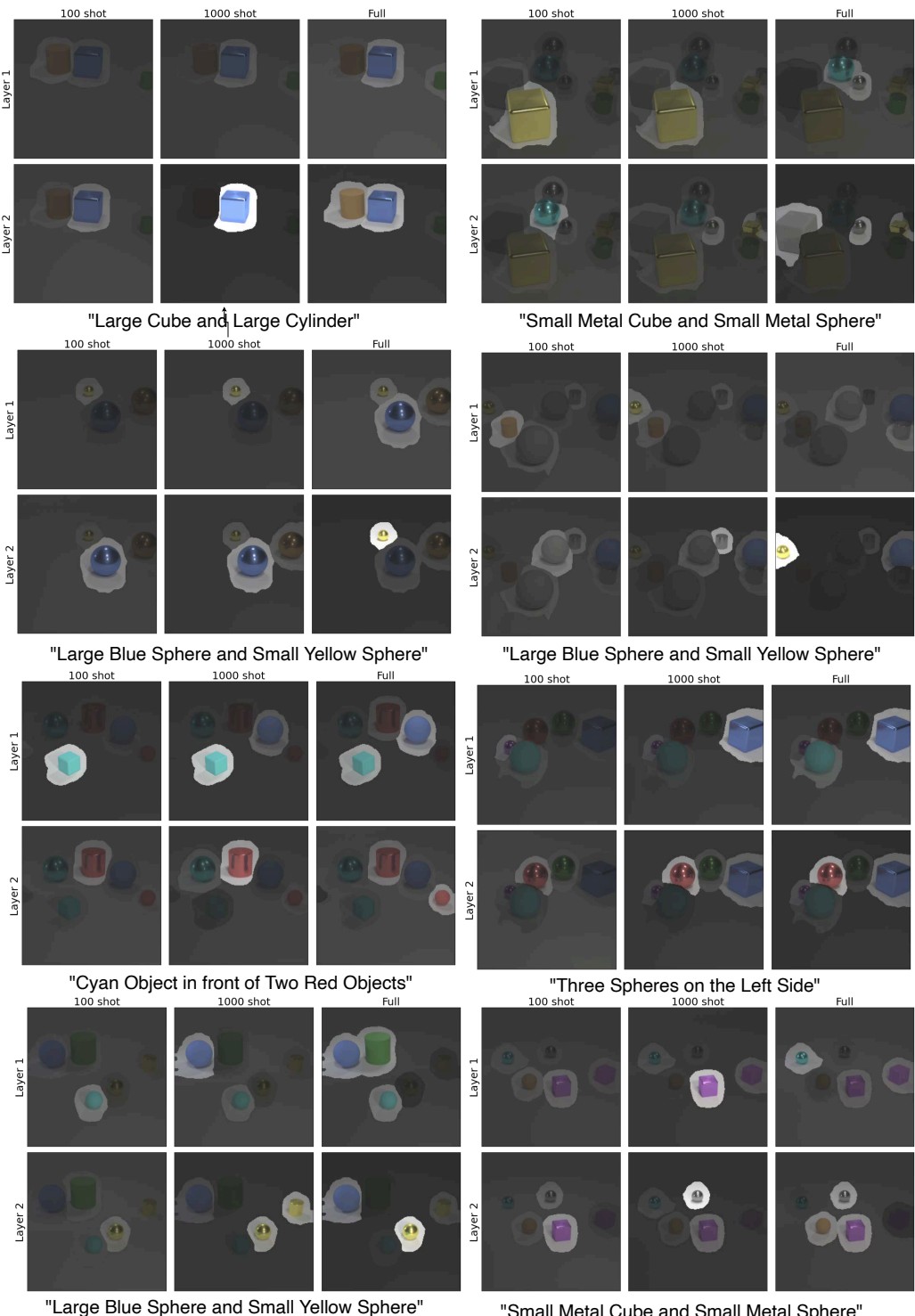

Figure 34: NSI visual rationales across different data regimes and attention depth.

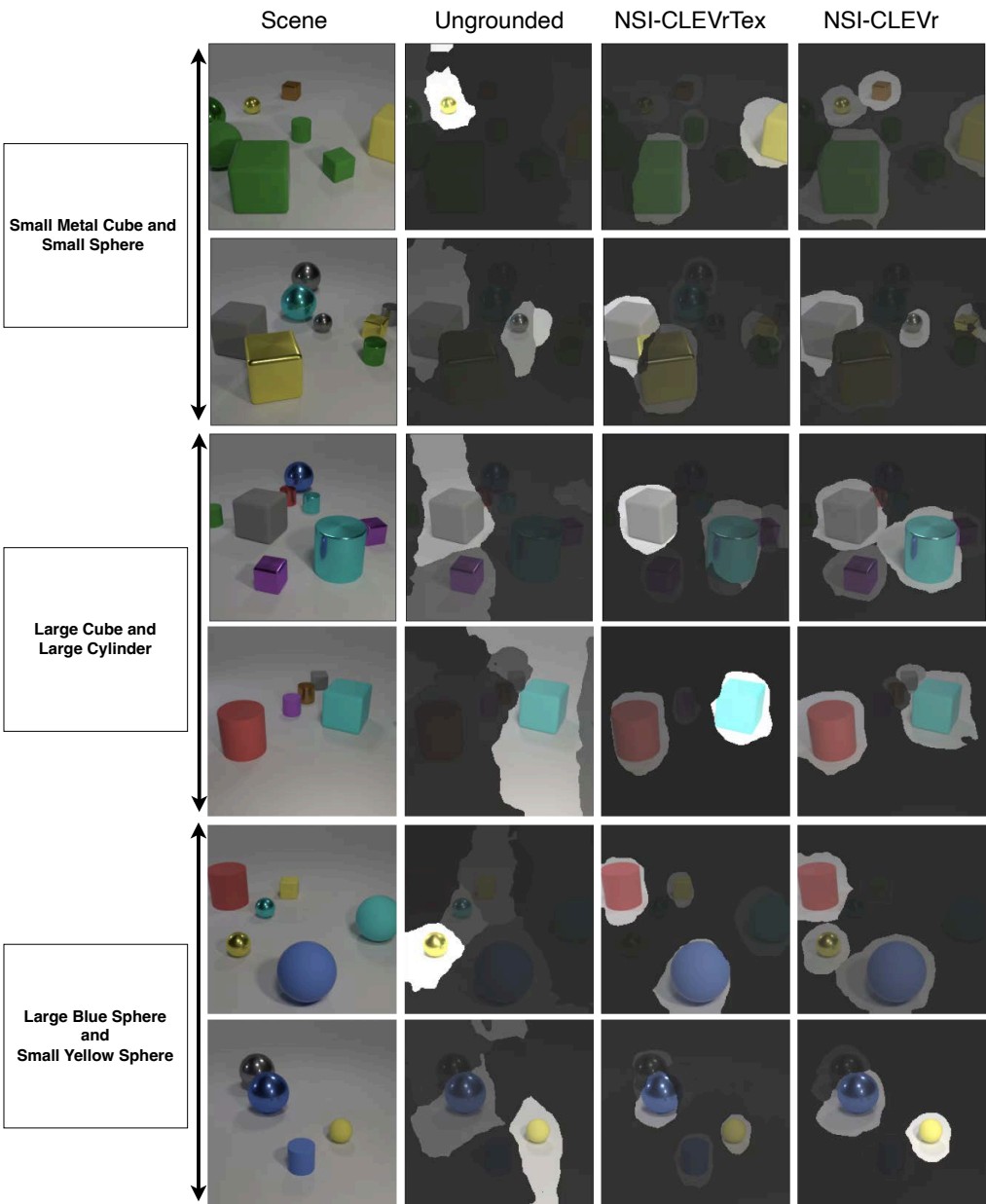

Figure 35: Visual rationales on Hans 3 across different methods.

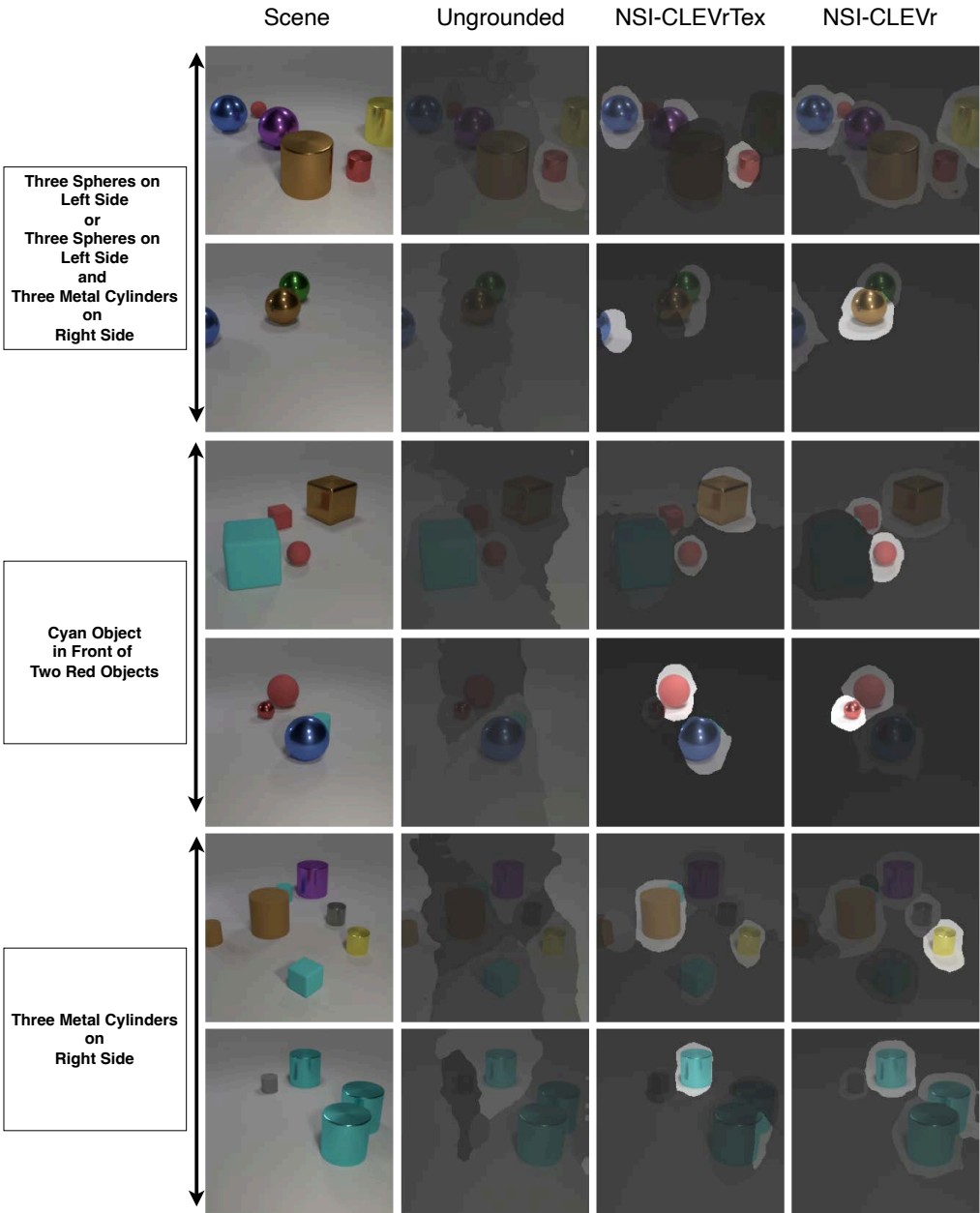

Figure 36: Visual rationales on Hans 7 across different methods.

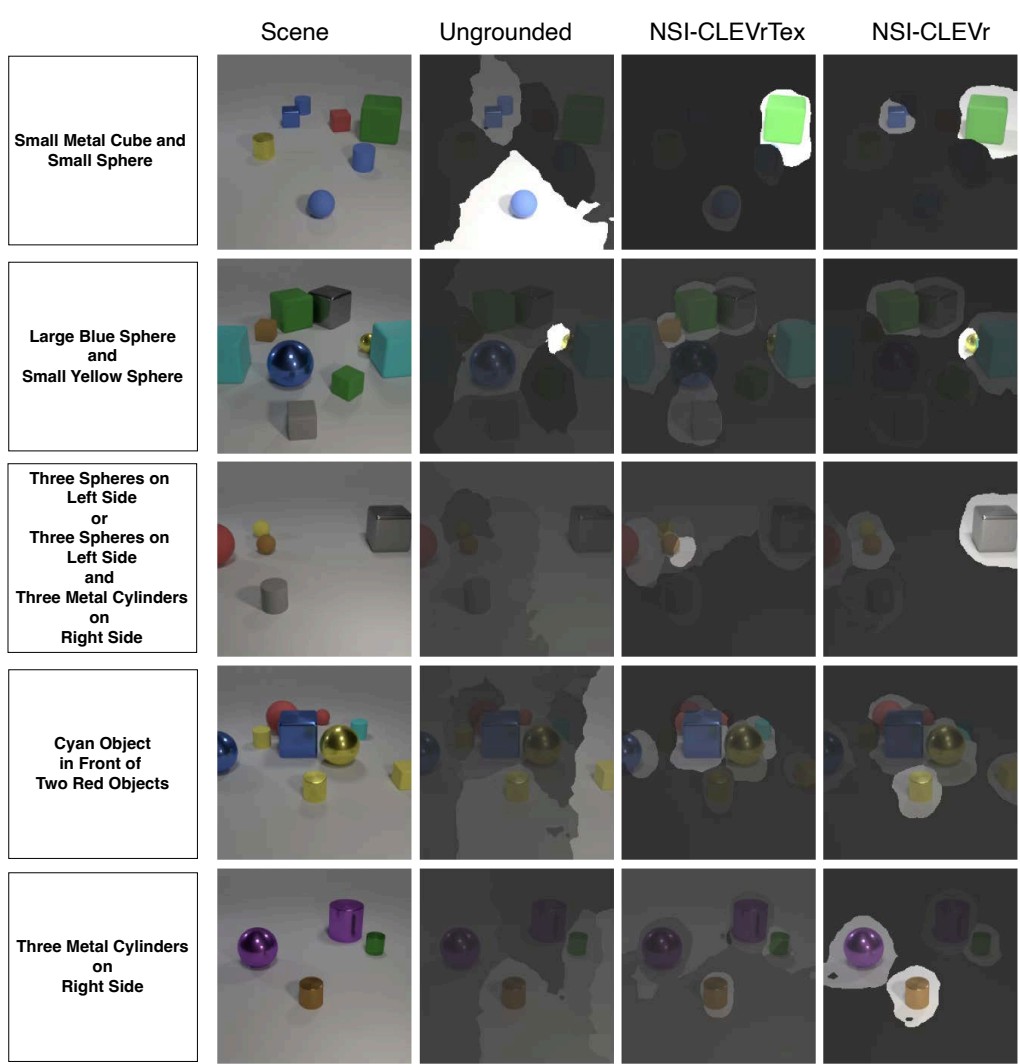

Figure 37: Some failure cases where NSI makes the correct prediction, albeit using incorrect or ambiguous support slots.

## F    OBJECT DETECTION WITH NSI

In this section, we demonstrate an architecture that uses inferred correspondences by NSI to perform object detection. We run preliminary experiments and explore the use of slots in real-world visual reasoning systems.

### F.1    NSI SCHEMA GENERATOR MODEL

We formulate the NSI schema generator (see Fig. 38) to predict, from each slot, its corresponding *visXML* primitives. To this end, we modify an encoder Transformer by interleaving cross-attention blocks with the self-attention layers to assimilate the context from the encoded slots. The input to the model is simply learned positional embeddings $\mathcal{P}^{1:N}$ that are decoded by $L$ attention-ensemble stacks to output primitive representations. We call this architecture $SET_\theta(.)$. The Slot Encoder Transformer (SET) is a stack of $L$ blocks, each computing (a) self-attention over inputs followed by (b) cross-attention of inputs over the context slot. Let $Z_{prim,l-1}^{dec,1:T}$ be the sequence representation at layer $l-1$ and $S$ denote the slot. Then

$$Z_{prim,L}^{dec,1:N} = SET_\theta(\mathcal{P}^{1:N}, S) \tag{13}$$

The cross-attention implementation of block $l$ is shown in Algorithm 2.

---

**Algorithm 2** Cross-Attention for Block $l$ of SET

---

**Require:** $Z_{prim,l}^{dec,1:T} \in \mathbb{R}^{T \times d}$, $T$ primitive embeddings from self-attention of layer $l$

**Require:** $\mathcal{S} \in \mathbb{R}^d$, Context Slot

Get query tokens: $Q_{l-1}^{1:T} = MLP_Q(Z_{prim,l}^{dec,1:T})$

Get keys, values of slot $\mathcal{S}$: $K, V = MLP_{KV}(S)$

Compute attention values: $M = \text{softmax}(Q^T K / \sqrt{d})$

Get output: $Z_{prim,l}^{dec,1:T} = M \times V$

---

Using an encoder-transformer-styled predictor has two advantages: (1) primitives can be generated in parallel and (2) each primitive representation is aware of the overall prediction context. $MLP$ property heads predict the object properties from the $Z_{prim,L}^{dec,1:N}$ representations. At each training iteration, the predicted properties $\hat{p}^{1:N}$ are optimally matched to their ground truth labels $p^{1:N}$ via an ordering $\sigma(1:N)$ obtained from Hungarian matching (HM) (Kuhn, 1955) on the property prediction loss $\mathcal{L}_{properties}$. Note that $\mathcal{L}_{properties}$ is a per-primitive loss obtained from the sum of individual property prediction losses. We use the cross-entropy loss for discrete properties, mean-squared error for continuous properties, and augment bounding-box regression with the Intersection over Union (IoU) loss. The training objective $\mathcal{L}_{gen}(.)$ is formulated as follows:

$$\sigma(1:N) = HM\left(\mathcal{L}_{properties}\left(p^{1:N}, \hat{p}^{1:N}\right)\right) \tag{14}$$

$$\mathcal{L}_{gen} = \sum_{i=1}^{N} \mathcal{L}_{properties}(p_i, \hat{p}_{\sigma(i)}) \tag{15}$$

At training time, we pad the aligned ground-truth instances with no-object labels $\Phi$ to account for representations without object predictions. In practice, bipartite matching for a single-slot instance is more computationally feasible than the overall image instance because per-slot object instances are significantly fewer.

### F.2    NSI SCHEMA GENERATOR HYPERPARAMETERS

The hyperparameters for the NSI schema generator are listed in Table 7.

### F.3    EXPERIMENTS

The correspondences from the train split are used to learn the NSI schema generator, as outlined in Appendix F.1. The schema generator decodes $T$ primitives from each slot, including the confidence

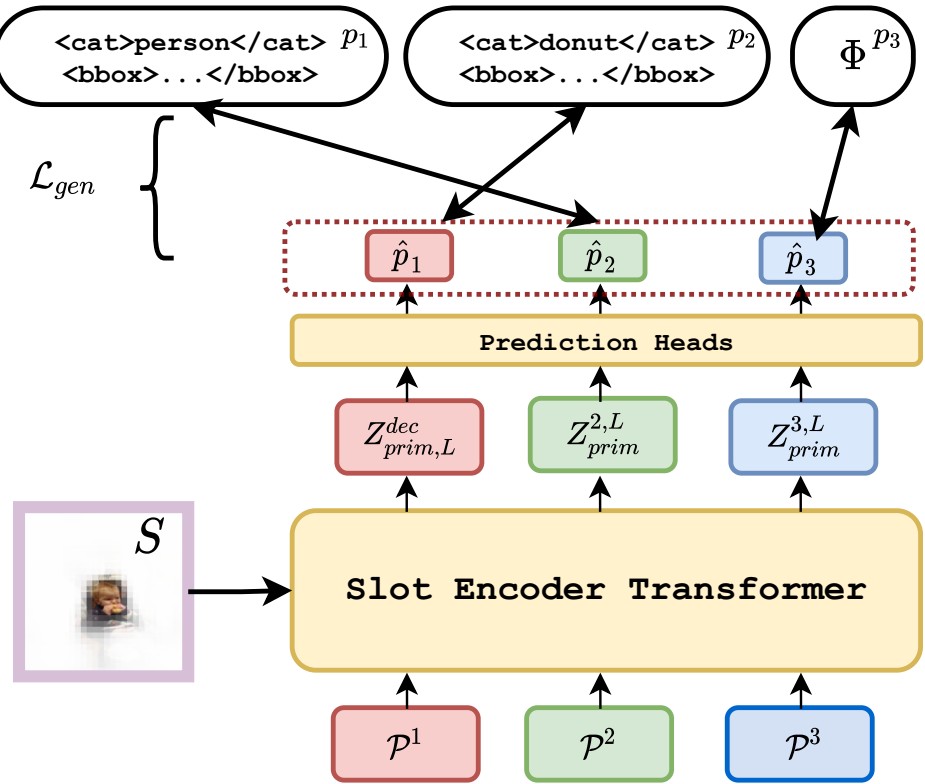

Figure 38: NSI schema generator model: A Slot Encoder Transformer attends to individual slots via cross attention and, in parallel, decodes tokens into schema primitives. At training time, a Hungarian set matching procedure assigns predictions to primitives associated with the slot. The prediction error is aggregated over assignments to compute $\mathcal{L}_{gen}$.

| Module | Hyperparameters | MOVi-C | COCO-10 | COCO-30 |
|---|---|---|---|---|
| SET | Num. Layers | 2 | 2 | 2 |
| | Num. Heads | 2 | 2 | 2 |
| | Hidden Dims. | 192 | 192 | 192 |
| | Predictions per Slot ($T$) | 5 | 8 | 8 |
| Inference | top-$M$ predictions | 10 | 10 | 30 |
| | Non-Max Suppression/Threshold | Yes (0.75) | Yes (0.75) | Yes (0.75) |
| Training Setup | Batch Size | 64 | 64 | 64 |
| | LR Warmup steps | 10000 | 30000 | 30000 |
| | Peak LR | $4 \times 10^{-4}$ | $1 \times 10^{-4}$ | $1 \times 10^{-4}$ |
| | Dropout | 0.1 | 0.1 | 0.1 |
| Training Cost | GPU Usage | 40 GB | 40 GB | 40 GB |
| | Days | 2 | 4 | 4 |

Table 7: Hyperparameters for the NSI schema generator model instantiation and training setup.

level of each object prediction. The overall schema for a single image is obtained as the top $M$ confident primitives out of predictions from all $K$ slots of that image. The NSI schema generator enables solving of downstream tasks with slots by making use of predicted properties from primitive tags. We demonstrate the usefulness of NSI for object detection.

**Object Detection Results:** In the context of real-world object detection on images, the *visXML* schema properties can be used to identify and locate objects in diverse scenes from the MOVi-C and COCO datasets. To this end, we extract the (`<cat>`) category and (`<bbox>`) bounding box fields from the generated primitives that depict the object category and location, respectively. For COCO, we test on two variants of the dataset: (1) **COCO-10**, a simple subset of the test split containing ten objects at a maximum, and (2) **COCO-30** that contains as many as 30 objects in a scene. We report the $AP_{IoU}$ metric across all three benchmarks. It denotes the area under the precision-recall curve for a certain IoU threshold (in %). We also run comparisons against the methods outlined in the retrieval experiments. Fig. 39 shows the experimental results and Figs. 40, 41 visualize object detection across various predictors. Our comparison baselines include:

(a) **Vanilla Slot Attention** (Locatello et al., 2020): The model is trained from scratch to resolve slot-object assignments through HMC.
(b) **DINOSAUR** (Locatello et al., 2020): It uses the DINO ViT backbone to learn slots for unsupervised object discovery by reconstructing perceptual features. The architecture is frozen while we train shallow predictors on top to detect objects.
(c) **DINOSAUR-FT**: We use the DINOSAUR model but fine-tune the architecture end-to-end on the prediction task.

We make the following observations:

(a) **NSI outperforms prior set-matching slot predictors**, especially by significant margins (20-30%) at lower IoU thresholds. Grounding object concepts in slots *a priori* improves the predictive power of slots. In comparison, matching the set of slots against the entire set of object annotations of the image yields poor generalization. In addition, large-scale pre-training and end-to-end fine-tuning are crucial ingredients, as evidenced by the subpar performance of the Vanilla and DINOSAUR methods on COCO.
(b) **The performance disparity between NSI and DINOSAUR-FT widens** as the complexity of scenes increases from COCO-10 to COCO-30. The inability of DINOSAUR to predict more than one object per slot necessitates modeling and learning to match up to 30 different slots, which generalizes poorly on novel scenes.
(c) **HMC slots deteriorate** for COCO-30 where the backbone is tasked with matching with 30 different objects at a time.

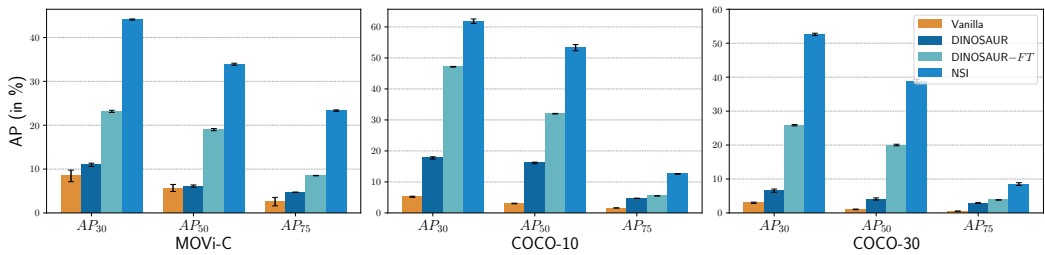

Figure 39: Object detection performance of various prediction methods on the MOVi-C, COCO-10, and COCO-30 benchmarks. We report $AP@IoU$ (higher is better) for different IoU thresholds. Standard deviation is reported over five random seeds.

**Slot Sweep on COCO Scenes:** Next, we investigate the effect of slot count on object detection. The number of slots is adjusted across a [5-30] range for the NSI schema generator and DINOSAUR-FT on the COCO-30 task. Fig. 42 presents the results of this experiment. For the DINOSAUR predictor, the number of slots imposes a strict upper limit on the number of object detections. As a result, the performance sharply deteriorates as the slot count is lowered. On the other hand, the NSI schema generator can detect multiple objects over a single slot and the performance drop is more graceful as the number of slots decreases.

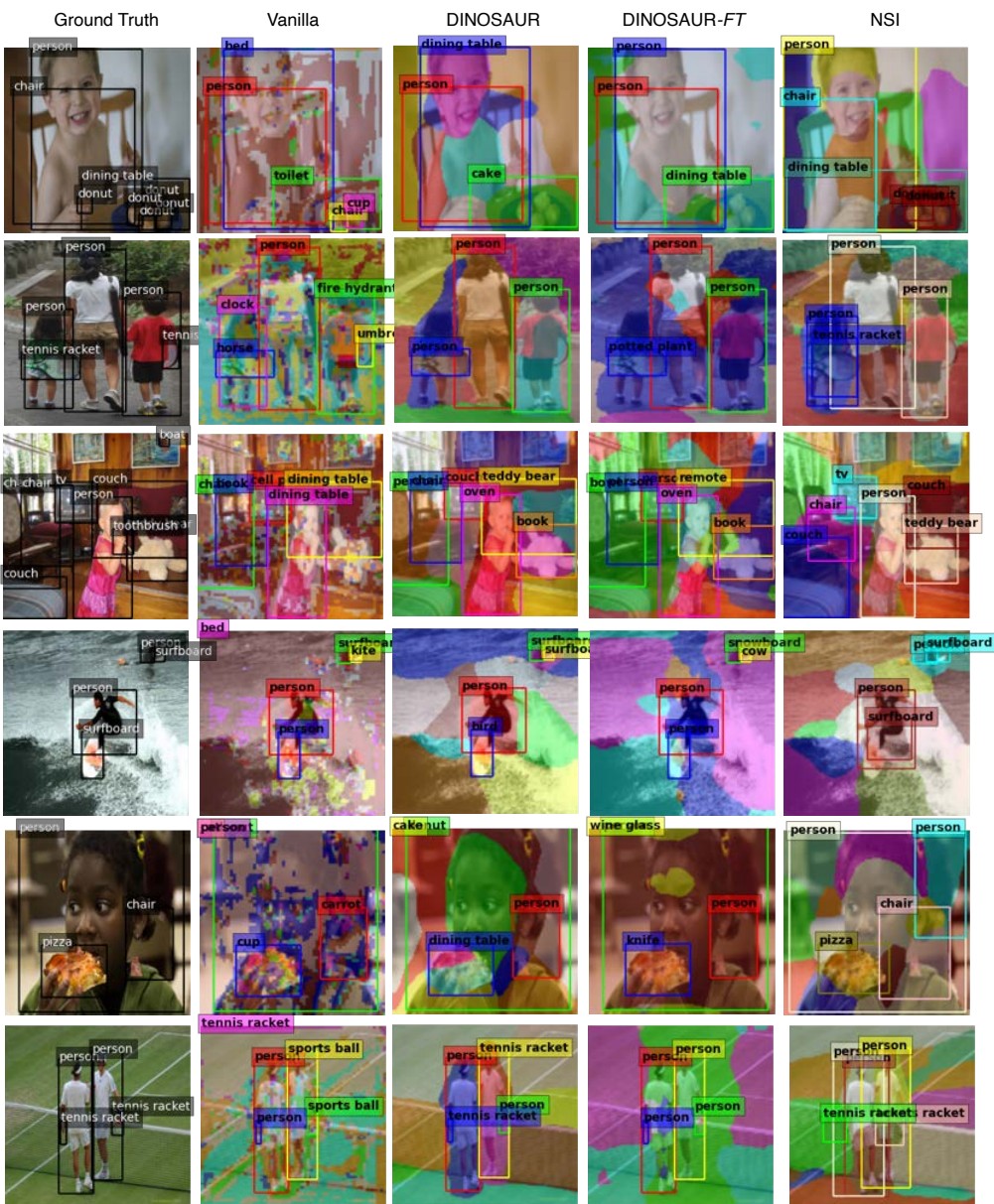

Figure 40: Object detection results on COCO scenes. NSI schema generator can flexibly detect multiple objects from the same slot, as evidenced by detections on COCO images. For example, a single slot predicts multiple 'donuts,' 'chair and dining table,' and 'person and tennis racket'.

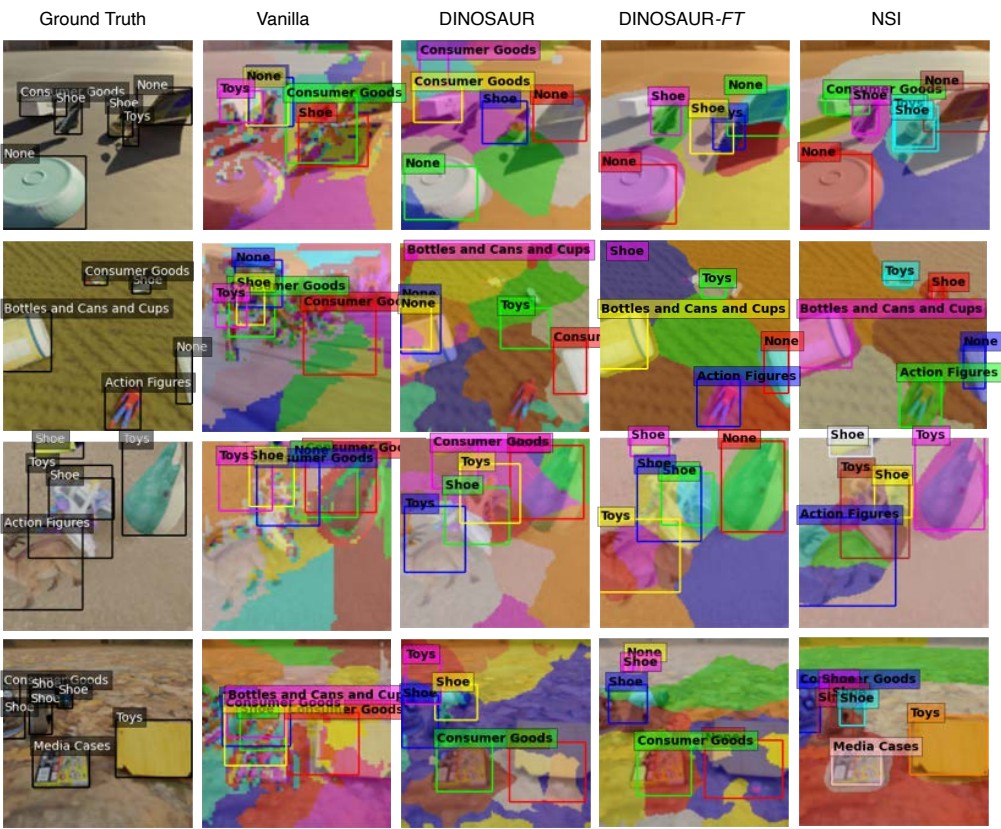

Figure 41: Object detection results on MOVi-C scenes.

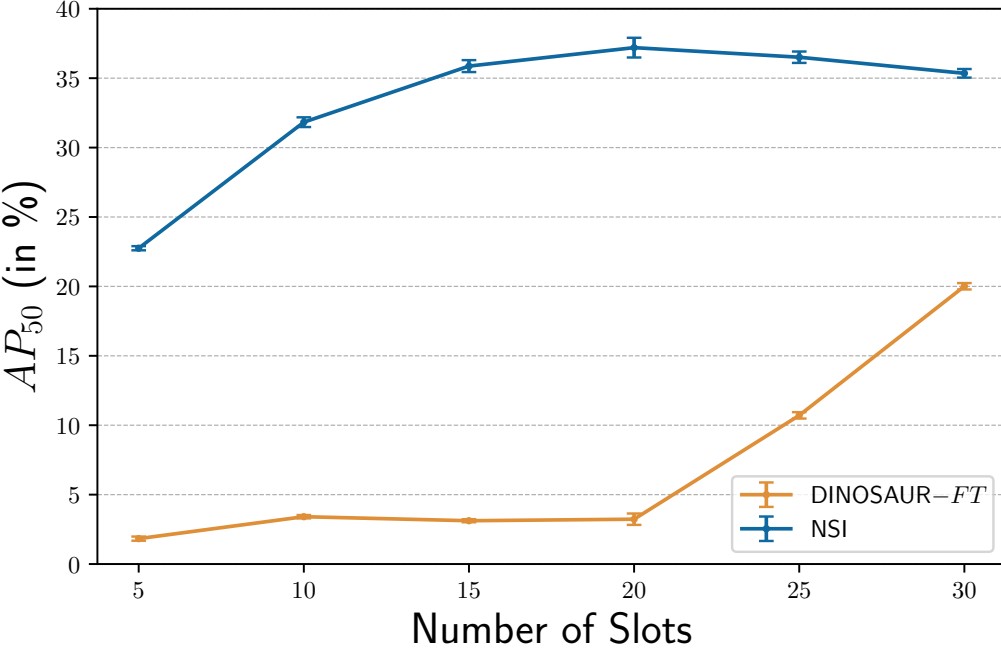

Figure 42: Effect of slot cardinality on object detection performance.

