# OpenReview forum: "Neural Slot Interpreters: Grounding Object Semantics in Emergent Slot Representations"
_ICLR.cc/2025/Conference — Submitted to ICLR 2025_

### Official Review · Reviewer_3CPV · 2024-10-29

**Soundness:** 3
**Presentation:** 2
**Contribution:** 1
**Rating:** 3
**Confidence:** 2

**Summary:**

This paper proposes neural slot interpreters, which provide an XML-like schema that uses simple syntax rules to organize the object semantics of a scene into object-centric schema primitives. It can learn to ground primitives into slots through an objective that reasons over the intermodal alignment. Experiments demonstrate that NSI tokenizers can achieve fewer tokens than other patch-based tokens.

**Strengths:**

1. This paper provides a contrastive learning objective that learns from image-schema pairs. This specific contrastive learning algorithm looks novel.

2. The proposed structured learning seems to benefit and outperform pure slot attention learning.

3. The experiment combining with vision transformers demonstrates slots as a better representation than patches.

**Weaknesses:**

1. The XML representation looks unnecessary, given a strong LLM model. Nowadays, multi-modal LLMs develop really fast, so as long as you have any testified representation, those language models can help you structure it into XML. In other words, the advantage of the proposed representation over the naive text representation (e.g., a caption model or a CLIP model) is very marginal.

2. The proposed experiments are limited in scale. CLEVR is too small to verify such an idea generally. The authors should experiment and report numbers on ImageNet. Also, ViT is not the SOTA, and it's interesting to see whether slot-based representations are better than patch-based ones.

3. Those slots are not accurately defined and it seems that those are just vector embeddings. Therefore, the reviewer doubts the motivation to research such slot-like representations.

4. The writing is not very clear. In fact, the methods are not explained very well (see questions below).

**Questions:**

1. What's the motivation for using two losses in Eq. 10? Those equations are introduced but not explained well. What's the connection of such a loss to contrastive losses used in prior works?

2. The bi-level architecture should be explained in the main text. What motivates learning the slot attention rather than learning a simple tokenizer? It seems that the authors wish to present a new tokenizer but it is neither simple nor scalable (to some other architecture).

3. What's the novelty of this work against general compositional learning works? The slot representations are not very well-defined and it's hard to claim novelty there.

---

> ### Author Response · Authors · 2024-11-22
>
> We thank the reviewer for the comments and constructive feedback. We respond to the questions and concerns below.
>
> >The XML representation looks unnecessary, given a strong LLM model. Nowadays, multi-modal LLMs develop really fast, so as long as you have any testified representation, those language models can help you structure it into XML.
>
> We agree with the reviewer. The key insight in this work is not about generating XML descriptions, which could indeed be done by LLMs, but rather using XML's hierarchical object-centric structure to ground visual concepts effectively. The paper shows that this structured organization enables better compositional grounding than language-only approaches (Section 4.3). The ease of generating and parsing XML descriptions using LLMs makes this approach even more practical and scalable.
>
> >What's the novelty of this work against general compositional learning works?
>
> While prior works [1,2,3,4] learn object-centric representations through reconstruction objectives alone, we propose a novel contrastive learning framework that explicitly grounds these representations in object semantics. Our approach differs fundamentally from existing compositional learning works in how we handle object-concept alignment. Prior works [1,4,5] use Hungarian matching (HMC) to enforce strict one-to-one mapping between slots and objects. Our NSI metric enables flexible many-to-many mappings between slots and object concepts, facilitates interpretability, and endows greater semantics to learned slots. Overall, our results show improved object discovery, grounding efficacy, and reasoning compared to both unsupervised slots and slots learned via HMC.
>
> >The proposed experiments are limited in scale. CLEVR is too small to verify such an idea generally. The authors should experiment and report numbers on ImageNet. Also, ViT is not the SOTA, and it's interesting to see whether slot-based representations are better than patch-based ones.
>
> While we agree that ImageNet is valuable for evaluating broad semantic categorization, CLEVr-Hans [6] was specifically chosen because it tests compositional reasoning capabilities in ways that ImageNet cannot. CLEVr Hans contains systematic confounders in the training set that require understanding object properties and their relationships beyond broad categories. Our goal was not to achieve SOTA performance but to investigate whether grounded slot representations can serve as more efficient tokens than patches. The ViT architecture was chosen as a controlled testbed for this comparison due to its simplicity, which helps isolate the impact of the tokenization strategy. Our future work aims to scale to real-world compositional reasoning tasks.
>
> >The slot representations are not very well-defined and it's hard to claim novelty there. Those slots are not accurately defined and it seems that those are just vector embeddings. Therefore, the reviewer doubts the motivation to research such slot-like representations.
>
> We build upon the vast established literature [1-5] of the Slot Attention mechanism (detailed in Section 3.2.1 and Appendix A.1), where slots emerge from an iterative attention process over visual features. While slots are vector embeddings, they are not arbitrary - they capture rich object-centric information. They can be used for downstream tasks like scene composition [7,8], learning intuitive physics [9],  and control [10]. Researching slot-like tokenizers is important because they can compress raw visual inputs into a tighter set of tokens endowed with object-centric semantics, facilitating compositional and efficient reasoning (see Section 4.4).
>
> >What's the motivation for using two losses in Eq. 10? Those equations are introduced but not explained well. What's the connection of such a loss to contrastive losses used in prior works?
>
> The two losses in Equation 10 serve complementary purposes in our approach:
> The reconstruction loss ($\beta_2$ × Lrecon) comes from traditional object-centric learning and ensures that slots can effectively encode and reconstruct the visual features of objects in the scene. This provides a foundation for learning spatially coherent object representations.
> The contrastive loss ($\beta_1$ × Lcontrastive) is our novel contribution that grounds semantic meaning into these object slots. It uses a symmetric cross-entropy formulation similar to prior contrastive learning works like SimCLR, but operates over object-property alignments rather than image-level features.
>
> As visualized in Figure 3,  the losses together capture spatial and semantic aspects of objects. We will make this connection more apparent in the final manuscript.
>
> >The bi-level architecture should be explained in the main text.
>
> Section 3.2.2 explains the bi-level architecture, where Equation 3 explains the lower-level primitive encoder snd Equation 4 describes the schema Transformer. We will make the motivation more clear in the final version.

---

> ### Author Response · Authors · 2024-11-22
> **.....continued**
>
> >It seems that the authors wish to present a new tokenizer but it is neither simple nor scalable (to some other architecture).
>
> NSI demonstrates strong scalability from synthetic datasets to complex real-world COCO scenes using just 15 tokens (vs hundreds in patch-based approaches), with Fig. 14 showing competitive segmentation performance even at small slot counts. The architecture is modular and can work with different backbones (demonstrated with DINO ViT and ResNet in our experiments) and bootstrap from strong pre-trained object-centric backbones while requiring minimal supervision (as shown in Fig. 7 where just 100 annotated examples achieve strong results). Additionally, NSI's semantic grounding improves object discovery compared to ungrounded baselines, with Fig. 5 showing better FG-ARI and mBO scores across CLEVrTex, MOVi-C and COCO datasets.
>
> [1] Object-Centric Learning with Slot Attention. Locatello et al. 2023
>
> [2] Illiterate DALL-E Learns to Compose. Singh et al. 2021
>
> [3] SAVi++: Towards End-to-End Object-Centric Learning from Real World Videos, Esayed et al. 2022
>
> [4] Bridging the Gap to Real-World Object-Centric Learning, Seitzer et al. 2023
>
> [5] Grounded Object Centric Learning, Kori et al. 2023
>
> [6] Right for the Right Concept: Revising Neuro-Symbolic Concepts by Interacting with their Explanations. Stammer et al. 2021
>
> [7] Object-Centric Slot Diffusion. Jiang et al. 2023
>
> [8] DORSal: Diffusion for Object-centric Representations of Scenes et al. Jabri et al. 2023
>
> [9] SlotFormer: Unsupervised Visual Dynamics Simulation with Object-Centric Models. Wu. et al. 2023
>
> [10] Neural Constraint Satisfaction: Hierarchical Abstraction for Combinatorial Generalization in Object Rearrangement. Chang et al. 2023

---

> > ### Comment · Reviewer_3CPV · 2024-11-22
> > **Thanks for the rebuttal.**
> >
> > It's not easy to rebuttal so many reviewers.
> > The reviewer acknowledges this rebuttal and final decision would be made after discussing with other peer reviewers.

---

### Official Review · Reviewer_Gvm1 · 2024-11-01

**Soundness:** 4
**Presentation:** 3
**Contribution:** 4
**Rating:** 8
**Confidence:** 5

**Summary:**

This work focuses on grounding slots – tokens that have a learned correspondence to objects in images. The authors build upon prior work, specifically Slot Attention and DINOSAUR. Their contribution is in using datasets of object properties to further ground slots computed with Slot Attention. They do so by proposing a novel contrastive loss. This loss pairs object representations based on object properties with object representations based on visual slots and then applies a standard InfoNCE objective over correct and incorrect pairings. The correct pairings are computed by greedy assignment based on the feature vector dot product. The model is evaluated in unsupervised object segmentation, object and scene retrieval and few-shot classification across synthetic and real-world datasets.

**Strengths:**

1. The authors propose to use visXML descriptions of objects and their properties to better ground object slots in Slot Attention. This is a novel combination of the spatial biases in Slot Attention and the semantic biases in object-centric annotations.
2. The authors propose a nearest neighbor slot to semantic embedding assignment mechanism that is shown to lead to better representations than hungarian matching. The key insight is to assign object properties to slots without enforcing an exclusive 1-to-1 mapping.
3. The paper includes a number of evaluations of the quality of the learned slot representation, including unsupervised segmentation, scene retrieval and few-shot classification. Each experiment has reasonable baselines.

**Weaknesses:**

1. It is not clear to me why the object properties get concatenated and processed with an MLP in Equation 3. Do we assume that each object has the same set of properties? Are there any missing properties? Some kind of permutation-invariant model (e.g., a Transformer or a graph neural network) would be more flexible with respect to these questions.
2. As far as I understand, the Transformer used in Equation 4 processes an entire scene. Doesn’t this result in a loss of distinction between individual objects to a certain extent, as we are presumably computing self-attention across all objects in the scene? I suppose this decision is already ablated in NSI-Schema Agnostic, which performs worse, but I am still curious about the rationale here. Would a per-object Transformer instead of a scene-level Transformer work?
3. The matching mechanism in Equation 6 might fail due to bad initialization, especially since Slot Attention is also sensitive to initialization. I would like to see a discussion of this.
4. The $S_{xx}, S_{xy}, S_{yx}$ notation in Equation 7 can be confusing.

**Questions:**

Points 1 and 2 in Weaknesses.

Given the recent improvements in token-based visual encoders, do we still need Slot Attention? Couldn’t we just use the tokens from ViT-CLIP or DINOv2, compress them into a smaller set of tokens using a cross-attention layer (like in Perceiver and PerceiverIO) and treat these tokens as slots? It is possible that your schema based loss makes the spatial biases that Slot Attention encodes redundant. So, your method could potentially be used without Slot Attention, which would overcome the scaling limitations and instabilities of SA.

---

> ### Author Response · Authors · 2024-11-22
>
> We thank the reviewer for the comments and positive feedback. We address the various concerns below.
>
> >It is not clear to me why the object properties get concatenated and processed with an MLP in Equation 3. Do we assume that each object has the same set of properties? Are there any missing properties? Some kind of permutation-invariant model (e.g., a Transformer or a graph neural network) would be more flexible with respect to these questions.
>
> We assume object annotations have all properties for this work and aim to explore missing properties in future work. The MLP was the most simple design choice, with graph neural networks being an improved design choice for enforcing permutation invariance on properties.  On the other hand, the schema encoder is permutation invariant to the annotated objects via the higher-level schema Transformer.  We have also explored missing objects in additional experiments presented in Appendix E2.5.
>
> >As far as I understand, the Transformer used in Equation 4 processes an entire scene. Doesn’t this result in a loss of distinction between individual objects to a certain extent, as we are presumably computing self-attention across all objects in the scene? I suppose this decision is already ablated in NSI-Schema Agnostic, which performs worse, but I am still curious about the rationale here. Would a per-object Transformer instead of a scene-level Transformer work?
>
> We proposed a bi-level schema encoder to prevent the loss of identity of objects compared to using the schema Transformer by itself.  The lower-level MLPs encode each object in isolation, agnostic to the other objects in the scene. Then the Transformer contextualizes all the scene objects into a schema-aware embedding.  As pointed out correctly, the schema-agnostic experiment explores encoding per-object embedding, and such an encoding, while assimilating object properties, cannot understand the relational abstractions between objects. We posit that a per object transformer could form a strong permutation invariant object encoder albeit still fail to understand the scene context.
>
> >The matching mechanism in Equation 6 might fail due to bad initialization, especially since Slot Attention is also sensitive to initialization. I would like to see a discussion of this.
>
> We agree and observe that the unsupervised objective helps break symmetry. In the final manuscript, we will include additional examples showing the absence of the reconstruction loss and how training solely on the contrastive loss leads to slot collapse.
>
> >Given the recent improvements in token-based visual encoders, do we still need Slot Attention? Couldn’t we just use the tokens from ViT-CLIP or DINOv2, compress them into a smaller set of tokens using a cross-attention layer (like in Perceiver and PerceiverIO) and treat these tokens as slots? It is possible that your schema based loss makes the spatial biases that Slot Attention encodes redundant. So, your method could potentially be used without Slot Attention, which would overcome the scaling limitations and instabilities of SA.
>
> We agree that using set encoder backbones like a Perceiver could potentially be performative in downstream reasoning tasks while circumventing the instability of slot attention. On the other hand, object slot representations have distinct compositional generalization properties that CLIP/DINO tokens don’t possess. Learning to ground object semantics has important implications for uncovering human-like systematic generalization and reasoning.

---

> > ### Comment · Reviewer_Gvm1 · 2024-11-25
> > **Response**
> >
> > Thank you for answering my questions! I am in favor of accepting the paper.

---

### Official Review · Reviewer_zw8T · 2024-11-03

**Soundness:** 2
**Presentation:** 2
**Contribution:** 2
**Rating:** 3
**Confidence:** 3

**Summary:**

The paper addresses the problem of grounding concepts to objects within images by training on paired images and object descriptions. It proposes an architecture built upon unsupervised object discovery models. During training, object features are first extracted by an object discovery module, and then these features are aligned with object property features through contrastive learning. Experiments demonstrate that the proposed method outperforms baseline models in object segmentation performance. Additionally, the learned representations benefit downstream applications, such as concept retrieval and concept reasoning.

**Strengths:**

1. The paper is easy to follow.
2. The paper includes the necessary details for reproducing the results.

**Weaknesses:**

- Using semantic grounding to enhance object discovery is not novel. For instance, [1] leverages question-answering and captions paired with images to improve the object discovery capabilities of Slot Attention. Author should also consider discussing other related works in concept grounding in images, such as [2] and [3].
- The experimental setting is somewhat unrealistic, as the paper assumes access to all objects and their attributes for every training image. In particular, the use of object position information (such as bounding boxes and 3D locations) during training significantly undermines the experimental results, as it contradicts the core motivation of object discovery. According to Figure 15, the proposed model performed worse than baseline models when trained without object position information.
- Referring to the semantic representation in the paper as VisXML is confusing, as there are no recursive structures involved. Only objects and their attributes are considered, which is not a substantial contribution.

[1] Language-Mediated, Object-Centric Representation Learning. Wang, et al. ACL Findings 2021.

[2] GroupViT: Semantic Segmentation Emerges from Text Supervision. Xu, et al. CVPR 2022.

[3] MDETR -- Modulated Detection for End-to-End Multi-Modal Understanding. Kamath et al. ICCV 2021.

**Questions:**

- Figure 7 shows only 100 grounding examples can have similar benefits on the downstream reasoning task compared with annotating the full dataset.  Does object discovery performance have the same trend?
- Can the model handle cases where only a subset of the objects in every image are annotated?

---

> ### Author Response · Authors · 2024-11-22
>
> >Figure 7 shows only 100 grounding examples can have similar benefits on the downstream reasoning task compared with annotating the full dataset. Does object discovery performance have the same trend?
>
> We ran this evaluation and results can be found in Appendix E1.4, Figure 16. The results were as follows (see manuscript for error bars):
>
> | # of Grounding examples | FG-ARI | $mBO^i$ | $mBO^c$ |
> |--------|-------|-------|-----|
> | Full | 44.24 | 28.1 | 32.1 |
> | 1000 | 43.32 | 27.75 | 31.94 |
> | 100 | 41.12 | 27.42 | 30.34 |
> | Ungrounded | 40.12 | 27.2 | 26.54 |
>
> Overall, we found that 100 annotations improved object discovery over the ungrounded baseline, especially for semantic segmentation ($mBO^c$ of 26.54 for ungrounded vs 30.34 for 100 grounded examples vs 32.10 for the full annotation)
>
> >Can the model handle cases where only a subset of the objects in every image are annotated?
>
> We conducted this evaluation in Appendix E2.5, Figure 30 where the maximum number of object annotations for MS COCO retrieval tasks were capped to 5 and 10 over two distinct settings. The results were as follows
>
> **Max. # of annotated objects: 10**
> | Model | Scene Retrieval |  | Property Retrieval |  |
> |-------|----------------|-----------------|-------------------|-----------------|
> |       | Recall@1 %| Recall@5 % | Recall@1 % | Recall@5 % |
> | CLIP | 31.28 | 52.45 | 27.82 | 48.32 |
> | HMC | 8.42 | 12.43 | 10.25 | 18.04 |
> | Resnet 34 | 3.41 | 11.49 | 5.32 | 15.02 |
> | NSI | 49.59 | 74.06 | 54.42 | 75.79 |
>
> **Max. # of annotated objects: 5**
> | Model | Scene Retrieval |  | Property Retrieval |  |
> |-------|----------------|-----------------|-------------------|-----------------|
> |       | Recall@1 % | Recall@5 % | Recall@1 % | Recall@5 % |
> | CLIP | 15.21 | 36.25 | 12.36 | 35.43 |
> | HMC | 4.21 | 8.73 | 5.45 | 11.12 |
> | Resnet 34 | 1.81 | 6.43 | 2.98 | 8.55 |
> | NSI | 29.64 | 51.7 | 32.73 | 52.73 |
>
> NSI was particularly resilient to the underspecified annotations showing the generalization of the nearest-neighbor metric. CLIP also showed robust generalization benefiting from its vast training corpora. In comparison, the Hungarian Matching grounding method, which is trained to match annotations to a single object, was less discriminative with the underspecification and struggled in these settings.
>
> >The experimental setting is somewhat unrealistic, as the paper assumes access to all objects and their attributes for every training image. In particular, the use of object position information (such as bounding boxes and 3D locations) during training significantly undermines the experimental results, as it contradicts the core motivation of object discovery. According to Figure 15, the proposed model performed worse than baseline models when trained without object position information.
>
> We politely disagree that positional information undermines our object discovery results, as both the HMC method and NSI have access to the same supervised signals - our contribution is in showing that NSI's contrastive alignment enables better utilization of whatever supervision is available. In addition, our experiments (Section 4.4, Appendix E1.4,  Appendix E2.5) show clear benefits from both partial annotations and fewer grounding examples. While we agree that the fully unsupervised setting is important, real-world applications often have access to at least partial object annotations, making our approach practically relevant.  Our ablation in Figure 15 shows that even without position information, NSI matches or outperforms HMC and ungrounded baselines on semantic metrics like $mBO^c$, indicating that our method effectively grounds object semantics regardless of spatial supervision.
>
> >Using semantic grounding to enhance object discovery is not novel. For instance, [1] leverages question-answering and captions paired with images to improve the object discovery capabilities of Slot Attention.
>
> We thank you for pointing us to relevant work and we will include a discussion on neuro symbolic methods and end-to-end training for concept grounding. We would also like to note that [1] relies on a neuro-symbolic parser with domain specific language containing primitives like ‘filter color’,’count objects’, ‘relate objects’ that are difficult to scale to real-world examples compared to scene annotations.
>
> >Referring to the semantic representation in the paper as VisXML is confusing, as there are no recursive structures involved. Only objects and their attributes are considered, which is not a substantial contribution.
>
> We agree that visXML is not a primary contribution. Our aim is to use the XML as an object centric organization  for contrastive slot grounding. While simple rather than recursive, this structure enables us to demonstrate clear gains in downstream tasks. We will revise the paper to properly frame visXML as a secondary contribution.

---

> > ### Comment · Reviewer_zw8T · 2024-11-27
> >
> > Thank the authors for the response. However, it does not address my concerns. Although baseline models receive the same object location signals as the proposed method, they are not designed to leverage such information. It makes more sense to directly train a supervised model for grounding (as reviewer V643 also mentions). If the authors claim their focus is “partial annotation” settings instead of fully supervised settings, they should compare with suitable baselines accordingly. Thus, I will keep my score.

---

> > > ### Author Response · Authors · 2024-11-27
> > >
> > > We thank the reviewer for their suggestions.
> > >
> > > > If the authors claim their focus is “partial annotation” settings instead of fully supervised settings, they should compare with suitable baselines accordingly.
> > >
> > > The Hungarian Matching baselines represent the current standard approach for supervised slot grounding. Our experiments show NSI outperforms Hungarian Matching across supervision levels (full, 1000, and 100 examples) and annotation settings.  We welcome specific recommendations for additional baselines appropriate for partial annotation setting if there are other relevant methods we should include.

---

> ### Comment · Reviewer_zw8T · 2024-11-27
>
> Given that object positions are available in the annotations during training, a highly related area of research is weakly supervised phrase grounding [1], where the aim is to ground textual phrases to image regions given image-caption pairs.
>
> [1] Contrastive Learning for Weakly Supervised Phrase Grounding. Gupta et al. ECCV 2020.

---

### Official Review · Reviewer_V643 · 2024-11-05

**Soundness:** 2
**Presentation:** 3
**Contribution:** 2
**Rating:** 3
**Confidence:** 5

**Summary:**

The paper presents the Neural Slot Interpreter (NSI), a model that aims to bridge the gap between visual perception and object-based reasoning by grounding object semantics in emergent slot representations. NSI organizes visual scenes using an XML-like schema (visXML) that assigns semantics to specific object-centered slots, allowing structured grounding through a contrastive learning objective.

**Strengths:**

The paper introduced a novel contrastive learning technique to align slot representations with visXML representations, which are essentially object-centric symbolic representations associated with the scene.
The evaluation was performed on a variety of tasks: (1) object discovery, (2) scene-property retrieval, (3) few-shot scene classification, and (4) object detection. The results demonstrate the efficacy of NSI.

**Weaknesses:**

The main weakness of the paper, to me, is the unclarity of the problem setting. VisXML, based on my understanding, is essentially some kind of symbolic scene representation similar to works such as http://nsd.csail.mit.edu/papers/nsd_cvpr.pdf I think, in order to obtain such kind of representations, one needs to train models with supervision on object masks/bounding boxes, and object labels (or maybe off-the-shelf models like large vision-language models or object segmentation models can already do the job). If that's the case, why do we still need to learn object-centric representations? Why don't we just directly supervise the learning of object segmentation etc. using these visXML labels...? I could imagine ways that combined supervised and unsupervised learning could help, but I don't think the paper shows sufficient evidence about their advantages.

Concretely, the paper should compare with a baseline that directly uses VisXML to do all these tasks.

Also, VisXML should not be listed as a contribution. See the NSD paper cited above.

**Questions:**

I don't have particular questions about the paper.

---

> ### Author Response · Authors · 2024-11-22
>
> We thank the reviewer for the comments and constructive feedback. We respond to the questions and concerns below.
>
> >The main weakness of the paper, to me, is the unclarity of the problem setting. VisXML, based on my understanding, is essentially some kind of symbolic scene representation similar to works such as http://nsd.csail.mit.edu/papers/nsd_cvpr.pdf I think, in order to obtain such kind of representations, one needs to train models with supervision on object masks/bounding boxes, and object labels (or maybe off-the-shelf models like large vision-language models or object segmentation models can already do the job).
>
> The core novelty of our work is not in the XML schema itself but in using it as a scaffold to learn grounded object-centric representations through contrastive learning. While VisXML and NSD use structured scene descriptions, our key contribution fundamentally differs. NSD and similar works aim to parse scenes into structured descriptions in a fully supervised way. Our goal is to ground object semantics into unsupervised object-centric learning via contrastive learning, not to perform inverse graphics or synthesize XML programs.
>
> >Why don't we just directly supervise the learning of object segmentation etc. using these visXML labels
>
> While this is the primary objective used by NSD and similar works to help program synthesis, it requires strong labels via segmentation masks and enforces strong one-to-one correspondences between predictions and ground truth. Rather than directly supervising object segmentation with VisXML labels, we use object properties as loose semantic guidance for slots where spatial masks naturally emerge through semantic alignment (Section 3.2, Section 4.2). Moreover, our contrastive metric allows flexible many-to-many mappings between slots and concepts, circumventing the overfitting from supervised learning due to rigid labeled correspondences.
>
> >Concretely, the paper should compare with a baseline that directly uses VisXML to do all these tasks.
>
> The HMC Matching baseline in our experiments (Figs. 5,6,7) directly uses VisXML properties to supervise slot predictions through Hungarian matching. This represents the straightforward approach of using VisXML for direct supervision. Our results show that NSI outperforms this direct supervision approach, particularly on:
>
> - Object discovery (Section 4.2, Fig. 5): NSI achieves 44.24% FG-ARI vs 32.18% for HMC on COCO
> - Property retrieval (Section 4.3, Fig. 6): NSI shows ~30-50% improvement in Recall@1/Recall@5 over HMC
> - Few-Shot classification (Section 4.4, Fig. 7): NSI shows ~3-20% improvement in classification accuration over HMC, in addition to generating interpretable visual rationaes

---

> > ### Comment · Reviewer_V643 · 2024-11-22
> >
> > Thanks for your response. Unfortunately, I think I am not convinced by the answers provided by the authors.
> >
> > First, "Rather than directly supervising object segmentation with VisXML labels, we use object properties as loose semantic guidance for slots where spatial masks naturally emerge through semantic alignment (Section 3.2, Section 4.2)." Yes, this is a very interesting experiment to do. However, I am not totally sure about the practical importance of this study. As I mentioned, if your training depends on VisXML, and your goal is to do tasks such as object segmentation, property retrieval or few-shot classification, you can directly supervise a model trying to predict the VisXML given the image, instead of using the proposed SlotAttention + VisXML alignment loss. That being said, I am looking for the practical advantage of the system, while agreeing on the soundness of the experiments. For the same reason, the HMC matching baseline is not really showing an important advantage of the system.
> >
> > Second, "The core novelty of our work is not in the XML schema itself." That sounds reasonable. Please lower the tone of this sentence in the introduction:
> >
> > Concretely, our contributions are as follows: 1. We present NSI, a co-training grounding paradigm for object-centric learners.  **We also propose an object-centric annotation schema called visXML for dense slot-label alignment. **

---

### Official Review · Reviewer_64BU · 2024-11-05

**Soundness:** 2
**Presentation:** 2
**Contribution:** 2
**Rating:** 6
**Confidence:** 3

**Summary:**

The current manuscript tackles the problem of grounding object-centric semantics in the slot representations that emerge via a visual reconstruction task. Towards this, it proposes a schema, called visXML, to represent the scene semantics in a structured form, and aligns it with the emergent slots via a contrastive loss within a given batch. The proposed approach, called Neural Slot Interpreter (NSI), results in a more grounded representation and outperforms some of the existing methods on bi-modal objective-property and scene retrieval tasks, in addition to outperforming patch-based tokens on few-shot classification tasks.

**Strengths:**

(S1) The problem tackled is interesting as it builds on top of existing object-centric slot learning to ground object concepts in them.

(S2) The proposed approach to use a visXML-based schema and measure scene-schema similarity is intuitive and simple to understand.

(S3) The manuscript is mostly well-written with sufficient motivation, relevant discussion of the literature, and clear technical details to understand the proposed approach, spread across the main and the appendix. I thoroughly enjoyed reading the manuscript.

**Weaknesses:**

(W1) Insufficient comparisons to prior work. The experiment tables do not report any numbers from the literature (e.g., [A, B, C, D]).
* In particular, [A] reports performance on slot discovery for MS COCO dataset as 35.27 (mBO_i) and 46.28 (mBO_c), while current approach reports 28.12 and 32.10 respectively.
* Further, if [C] is the Ungrounded baseline, the numbers do not match with those shown in the literature, mBO_c of 31.1 [C] vs 26.5 (reported in Appendix E.1.1). Request the authors to include necessary comparisons and fix these inconsistencies.

(W2) Limited experimentation on real world datasets. Though the results on synthetic datasets like CLEVr Hans, CLEVrTex, and MOVi-C motivate the problem, the real-world applications are more guided by experimentation on real world datasets. This also helps understand the efficiency of the approach beyond the sim2real domain difference. While the paper includes results on MS COCO, results on PASCAL VOC (often reported in prior works in the area) are missing.

Without these results, the quantitative benefits of the proposed approach are unclear.

(W3) A minor concern is the lack of relevant pointers in the main, when referring to modules from prior work. For instance, it is unclear how slots are obtained in Sec 3.2.1 without reference to either the appendix A.1 or prior work. Request the authors to go through the manuscript carefully to provide these details.

**References:**
* [A] Guided Latent Slot Diffusion for Object-Centric Learning. https://arxiv.org/pdf/2407.17929
* [B] Object-Centric Slot Diffusion. https://arxiv.org/pdf/2303.10834
* [C] Bridging The Gap To Real-World Object Centric Learning. https://arxiv.org/pdf/2209.14860
* [D] SlotDiffusion: Object-Centric Generative Modeling with Diffusion Models. https://arxiv.org/pdf/2305.11281

**Questions:**

Request the authors to address the weaknesses to demonstrate the quantitative benefits of the current approach compared to all existing methods (even if it does not have superior performance).

---

> ### Author Response · Authors · 2024-11-22
>
> We thank the reviewer for the comments and constructive feedback. We respond to the questions and concerns below.
>
> >Limited experimentation on real world datasets. Though the results on synthetic datasets like CLEVr Hans, CLEVrTex, and MOVi-C motivate the problem, the real-world applications are more guided by experimentation on real world datasets. This also helps understand the efficiency of the approach beyond the sim2real domain difference. While the paper includes results on MS COCO, results on PASCAL VOC (often reported in prior works in the area) are missing.
>
> Our initial focus on MS-COCO was motivated by its greater complexity and challenge (MS-COCO: 91 object categories, averaging 7.7 instances per scene vs Pascal VOC: 20 object categories, averaging 2.3 instances per scene)
>
> - **Zero Shot Pasal VOC Evaluation:** To address this feedback while meeting rebuttal time constraints, we evaluated zero-shot transfer to Pascal VOC using our COCO-trained models (Appendix E1.1, Figure 20). This setup tests real-world scene understanding and also evaluates out-of-distribution generalization
>
> - **Results:** Our experiments show that NSI's benefits generalize well to Pascal VOC, with consistent improvements over the ungrounded baseline even with zero-shot transfer.
>     1. FG-ARI: +1.54 % relative improvement
>     2. $mBO^i$:  +1% relative improvement
>     3. $mBO^c$: +1.12% relative improvement
>
> >Further, if [C] is the Ungrounded baseline, the numbers do not match with those shown in the literature, mBO_c of 31.1 [C] vs 26.5 (reported in Appendix E.1.1). Request the authors to include necessary comparisons and fix these inconsistencies.
>
> We appreciate the reviewer's careful attention to the comparison with DINOSAUR [C]. While our work builds on the DINOSAUR recipe, there are important implementation differences that explain the variation in reported numbers:
>
> - **Encoder Architecture:** While [C] uses DINO ViT B-16 as the encoder, our implementation uses DINO ViT B-8 with a smaller patch size.
> - **Decoder Architecture:** While [C] uses 4 MLP layers with 2048 hidden dimensions, we use 3 MLP layers with 1024 hidden dimensions
> - **Experiment control:** To isolate the effect of NSI grounding, we maintained consistent hyperparameters across all compared methods (ungrounded, HMC matching, and NSI) and controlled for computational budget. This allows for a fair comparison of the grounding techniques while acknowledging that absolute performance numbers may differ from [C] due to the architectural choices above
>
> We will update the paper to explicitly clarify these implementation differences and ensure all comparisons are clearly contextualized to [C].
>
> >Insufficient comparisons to prior work. The experiment tables do not report any numbers from the literature (e.g., [A, B, C, D]).In particular, [A] reports performance on slot discovery for MS COCO dataset as 35.27 (mBO_i) and 46.28 (mBO_c), while current approach reports 28.12 and 32.10 respectively.
>
> We thank the reviewer for highlighting these comparisons. While computationally intensive diffusion-based methods [A,B,D] achieve superior absolute performance, the primary goal of NSI was to demonstrate the benefits of explicit semantic grounding using widely adopted lightweight backbones like [C] that have proven real-world generalization capabilities. We believe NSI's grounding mechanism is complementary to state-of-the-art decoders - the paradigm we demonstrate could potentially enhance more sophisticated architectures while maintaining their computational advantages. Due to the time constraints of the rebuttal period, we will report the difference between the state-of-the-art experiments and show preliminary grounding experiments on [B]/[D]  in the final version.
>
> >A minor concern is the lack of relevant pointers in the main, when referring to modules from prior work. For instance, it is unclear how slots are obtained in Sec 3.2.1 without reference to either the appendix A.1 or prior work. Request the authors to go through the manuscript carefully to provide these details.
>
> We have included implementational details in Section 4.1 but we will carefully integrate these references in the methods section.

---

> ### Comment · Reviewer_64BU · 2024-11-27
> **Post-Rebuttal Discussion**
>
> Thanks to the authors for their response and carefully addressing my concerns. As a result, I am raising my rating to reflect that my vote is to accept the publication in the current form.

---

### Author Response · Authors · 2024-11-22
**Manuscript Updates and clarification**

We thank all reviewers for raising several great questions, adding detailed comments, and drawing our attention to related works. We will incorporate the feedback in the next iteration of the manuscript. Below, we address some common concerns and summarize additional results in the manuscript.

**Additional experiments**

Based on the feedback, we have updated our manuscript with the following experiments:

- Zero-Shot Pascal VOC 2012 object discovery experiments (Appendix E.1.1 and Figure 20)
- Ablation on the number of grounding examples for object discovery on MS COCO (Appendix E1.4 and Figure 16)
- Grounding with underspecified object annotations (Appendix E2.5 and Figure 30)

Additions to the original manuscript can be found highlighted in red text.

**Clarification about novelty and use of VisXML**

We acknowledge and agree with the reviewers V643, zw8T and 3CPV that  visXML by itself should not be considered a primary technical contribution, as similar markup languages have been used in prior work like NSD [1]. While prior works focus on synthesizing markup languages (eg: inverse graphics) we use XML to organize already available annotations into object-centric hierarchies. The key novelty of our work lies in using this schema to synergize object-centric representations with object-centric semantics without the need of strong one-to-one labels or semantic mask supervision.

[1] http://nsd.csail.mit.edu/

---

### Meta-Review · Area_Chair_X9YY · 2024-12-18

**Metareview:**

This paper proposes a method (NSI) for object-centric learning from visual data, while grounding this using an XML schema (which the authors called visXML). The method uses DINOSAUR (slot-attention based backbone) additionally trained on scene-schema pairs using contrastive loss. The authors perform experiments on CLEVr Hans, CLEVrTex, MOVi-C, and COCO, on tasks including object discovery, few-shot classification, and scene property retrieval. NSI is compared to ungrounded slots, as well as a baseline using Hungarian Matching Criterion (to use the visXML as supervision for slot predictions) to show improvements in performance. While the results in the paper is promising, and represents an interesting application of contrastive loss for grounding properties in object-centric learning, concerns are raised by reviewers on the choice of baselines and experimental setup. The main concern raised by reviewers is that the proposed approach requires annotations and supervision to ground the slots, and reviewers did not consider HMC to be a sufficient baseline for comparison (e.g. one could directly train a model to predict these properties; weakly-supervised grounding was brought up by one of the reviewers). The AC agrees that in its current form, with the level of annotations required for NSI, comparison and including a discussion of this work in relation to weakly supervised approaches would be valuable + experiments in the main paper on varying levels of annotations, would better demonstrate the significance of this approach. I encourage the authors to take into account helpful comments raised by reviewers with remaining concerns and I think this work could definitely make a strong paper with these additional discussions and experiments.

**Additional Comments On Reviewer Discussion:**

The reviewers initially raised concerns about whether VisXML is a technical contribution itself, some concerns on experiments, and clarification questions (e.g. on the losses). The authors responded by adding additional experiments results (e.g. clarifying differences with DINOSAUR, adding PASCAL VOC as an additional dataset in object discovery etc.), clarifying the use of the HMC baseline, clarifying that visXML is not the main technical contribution, and answered clarification questions well. Some reviewer concerns were addressed at this stage. A subset of reviewers participated in the AC-reviewer discussion after the rebuttal, and agreed on the rejection decision, mainly based on the concern of the additional supervision required and suggestions for additional baselines and experiments that also use those supervision.

---

### Decision · Program_Chairs · 2025-01-22

Reject